# Beyond Open-World: COSRA, a Training-Free Self-Refining Approach to Open-Ended Object Detection

## Abstract

Traditional object detection models rely on predefined categories, which limit their ability to recognize unseen objects in open-world settings. Recent efforts in open-world and open-ended detection have begun to address this challenge by enabling models to go beyond closed-set assumptions. However, these approaches often remain limited in terms of scalability, adaptability, or generalization to diverse environments. To overcome these restrictions, we introduce a Context-oriented Open-ended Self-Refining Annotation model (COSRA), a training-free framework that combines context-aware reasoning with self-learning for open-ended object labeling. COSRA leverages Large Language Models (LLMs) to generate candidate labels for unknown objects based on contextual cues from known entities within a scene. COSRA then pairs these labels with visual embeddings to construct an Embedding-Label Repository (ELR), enabling inference without category supervision. To further enhance consistency, we introduce a self-refinement loop that re-evaluates repository labels using visual cohesion analysis and KNN-based majority relabeling. For a newly encountered unknown object, COSRA retrieves visually similar instances from the ELR and applies frequency-based voting and cross-modal re-ranking to assign a robust label. Our experimental results on COCO and LVIS datasets demonstrate that COSRA outperforms state-of-the-art methods and effectively annotates novel categories using only visual and contextual signals without requiring any fine-tuning or retraining.

## 1 Introduction

Imagine you are sitting in a restaurant in a foreign country. On the table are several familiar items—forks, bread, napkins, a salt shaker—but there are also a few objects you have never seen before. Since no one has told you their names, you cannot identify them at first. Later, when visiting another restaurant or dining at someone's home, you notice similar and yet unknown object again. From its repeated presence at meal settings, you infer that it must be associated with dining. By examining its appearance—whether it is a liquid, solid, or gel—you further hypothesize that it may be food or a condiment. Over time, you recall that in conversations around these occasions, the same word was often spoken whenever this object appeared. Combining this contextual association, your observations of its properties, and the recurring word, you form a strong hypothesis about its name. In summary, the process unfolds as follows: first, recognizing visually similar objects in similar contexts; next, considering its visual features to infer possible affordances; and finally, noticing the recurrence of words associated with the object, which enables it to be named. This principle is central to the COSRA methodology: open-world learning from context. COSRA aims to mimic this human capability—the learning of previously unfamiliar entities in an open-world environment.

Modern object detection models have achieved considerable success in identifying objects from predefined categories in controlled environments (Lin et al., 2024a; Cai & Vasconcelos, 2018; Carion et al., 2020; He et al., 2017; Ren et al., 2016; Sun et al., 2021). However, their utility remains confined to closed-set settings, where all test-time categories are known at training time. These models fundamentally rely on supervised learning over fixed label spaces, rendering them ineffective in open-world environments where novel or long-tail object categories frequently appear. Recent advances in open-world recognition attempt to address these limitations by leveraging large-scale

vision-language models (VLMs) (Radford et al., 2021b; Li et al., 2022b), prompt-driven classification (Gu et al., 2022), or knowledge-enhanced object detectors (Zang et al., 2022) to detect wider varieties of objects. While these models have made progress in various aspects of object recognition or discovery, no existing system fully solves open-ended *naming* cleanly—i.e., given a truly unseen object, produce a correct semantic label (e.g., "giraffe") without that name ever appearing in its vocabulary. In practice, methods either detect novelty and return a generic *unknown* tag (Joseph et al., 2021; Liang et al., 2023), or cluster novel instances into surrogate buckets (e.g., *unknown-1*) (Wu et al., 2022) rather than providing an actual name. This gap motivates our training-free, context-driven approach.

To address this challenge, we introduce a Context-oriented Open-ended Self-Refining Annotation model (**COSRA**): a *training-free*, *context-aware reasoning* and *self-learning* framework that can assign appropriate semantic names to previously unseen objects in open-world, open-vocabulary settings. COSRA represents a paradigm shift in object detection by moving beyond fixed, category-constrained supervision and instead emulating the human-like ability to reason and infer object categories from context. To achieve this, COSRA builds an **Embedding-Label Repository (ELR)** directly from data, without relying on gradient-based updates or task-specific retraining. When a pretrained detector encounters an unknown object, COSRA formulates a query to a large language model (LLM), asking for possible identities of the object while providing contextual information such as the identities, locations, and sizes of nearby objects, along with visual attributes of the unknown object (e.g., color, texture). The LLM generates candidate labels, which are associated with the object's visual embeddings and stored in the ELR. Once the ELR contains a sufficient number of elements, COSRA measures visual similarity to find each element's $k$-nearest neighbors and applies majority voting over their labels to produce a shortlist of candidate labels for visually similar objects.

Our key contributions are as follows:

- We propose a novel zero-shot framework for open-ended object labeling that bypasses the need for supervised training on unknown categories.
- We design a context-aware prompt construction mechanism that integrates visual descriptors and spatial layouts to effectively guide large language models.
- We introduce an embedding-label memory module and a two-stage refinement process combining frequency-based aggregation and cross-modal re-ranking to improve robustness and disambiguate noisy predictions.
- Experiments on COCO and LVIS highlight COSRA's ability to annotate novel object categories from contextual and visual signals, establishing new performance benchmarks without requiring backbone fine-tuning.

## 2 RELATED WORK

**Open-Vocabulary Object Detection.** The advent of vision–language models (VLMs) has shifted detection from category-supervised setups to Open-Vocabulary Object Detection (OVD) that exploits image–text pretraining. Many OVD methods classify region features against a user-provided label list at inference, typically via CLIP-style similarity (Zareian et al., 2021; Radford et al., 2021a). SAMP (Zhao et al., 2024) strengthens this paradigm by learning scene-adaptive prompts and region-aware visual–text alignment to better adapt CLIP for detection. Nevertheless, state-of-the-art systems—including OWL-ViT (Minderer et al., 2022), GLIP (Li et al., 2022b), Detic (Zhou et al., 2022), and YOLO-World (Cheng et al., 2024)—still require anticipating which categories may appear and supplying long candidate vocabularies, leaving OVD fundamentally conditioned on a closed set at test time.

**Open-World Object Detection.** To circumvent reliance on fixed label lists, Open World Object Detection (OWOD) methods, such as ORE (Joseph et al., 2021) and UnSniffer (Liang et al., 2023), have been introduced. ORE leverages contrastive clustering and an energy-based unknown identifier to flag novel instances for incremental learning, while UnSniffer adopts a generalized object confidence score with graph-based box selection. However, both approaches ultimately rely on human annotation to incorporate new classes. Transformer-based works such as (Maaz et al., 2021), OW-DETR (Gupta et al., 2022), and PROB (Zohar et al., 2023) aim to identify unknowns but do not predict labels, and OSODD (Zheng et al., 2022) clusters unseen classes into novel groups without

naming them. While these approaches advance unknown object detection and classification under provided label sets, none are capable of generating novel labels for unseen classes without oracle supervision.

**Open-Ended Object Detection.** Open-Ended Object Detection (OED) addresses the challenge of detecting and labeling objects without relying on a fixed vocabulary, allowing models to assign meaningful names to both known and previously unseen categories. GenerateU (Lin et al., 2024a) first introduced this problem by leveraging Deformable-DETR (Zhu et al., 2020) as a region proposal generator, whose outputs are passed to a language model for label assignment. To address Generate-U's dependency on large-scale data, slow convergence, and limited accuracy, Open-Det (Cao et al., 2025) introduces a vision–language aligner with prompt distillation, and employs improved loss functions, achieving stronger performance with less data and computation. DetCLIPv3 (Yao et al., 2024) extends open-vocabulary detection by integrating a caption head for hierarchical label generation, supported by large-scale auto-annotated image–text pairs and an efficient two-stage training strategy. Unlike these training-based approaches, VL-SAM (Lin et al., 2024b) uses a training-free strategy by coupling vision–language models with the Segment Anything Model (SAM) (Kirillov et al., 2023), using attention-based prompts to discover and segment unseen objects without predefined categories.

**Context-Aware Reasoning.** Early open-vocabulary detection methods typically rely on exhaustive class lists, where models such as CLIP (Radford et al., 2021a) or RAM (Zhang et al., 2023) match image features to pre-defined text labels. More recent work aims to overcome this limitation by leveraging large language models (LLMs). For example, RAM++ (Huang et al., 2023) enriches class tags with additional descriptors derived from ground-truth annotations, while DVDet (Jin et al., 2024) introduces fine-grained text descriptors of object parts to improve matching with visual embeddings. Other studies similarly exploit textual cues, such as texture information (Wu & Maji, 2022) or general descriptors in OvarNet (Chen et al., 2023), showing that richer text features can indirectly facilitate novel label generation. A parallel line of research integrates image and language more holistically. Multi-modal models such as BLIP (Li et al., 2022a) and LLaVa (Liu et al., 2023a) can generate free-form captions without relying on class lists, but their performance hinges on massive supervised datasets. More recently, contextual reasoning with LLMs has emerged as a promising direction. LaMI-DETR (Du et al., 2024) addresses limitations of CLIP-based OVD by using language model instructions to refine concept representations and mitigate base-category bias. More recently, LLaMA (Touvron et al., 2023) has been applied for contextual reasoning, enabling the generation of contextually aware labels that enhance detection under occlusion and poor visibility (Rouhi et al., 2025).

## 3 METHODOLOGY

We present COSRA, a framework for open-ended object detection that integrates multi-modal foundation models with contextual reasoning. Unlike conventional open-world approaches that only flag novel objects as "unknown," COSRA advances further by acting as an open-ended framework, i.e., it assigns semantic labels to previously unseen categories without requiring training or manual supervision. We provide the theoretical motivation for COSRA, grounded in conditional entropy, in Appendix A.

COSRA operates in three sequential stages: (1) **Unknown object detection**, (2) **Embedding–Label Repository (ELR) construction**, and (3) **Iterative refinement**. In the first stage, we use the Segment Anything Model (SAM) (Kirillov et al., 2023) and Faster R-CNN (Ren et al., 2016) to separate known from novel objects. In the second stage, we integrate context-aware characterization, LLM-driven candidate label generation, and cross-modal re-ranking with CLIP to link embeddings with semantic hypotheses. In the final stage, we improve noisy assignments through neighborhood voting and visual cohesion analysis. See Figure 1.

### 3.1 UNKNOWN OBJECT DETECTION

We adopt a two-step process to distinguish between known and candidate unknown objects. First, SAM generates class-agnostic region proposals $\mathcal{R} = \{r_1, \ldots, r_i, \ldots, r_N\}$ for an input image $\boldsymbol{I} \in \mathbb{R}^{H \times W \times 3}$, ensuring broad coverage of all salient regions. Second, a detector pretrained only on

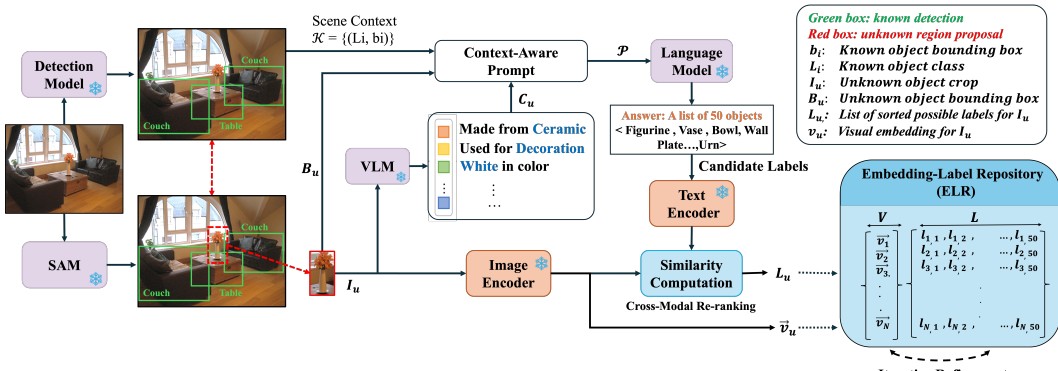

**Figure 1: Overview of the COSRA framework.** It consists of three sequential stages: (1) unknown object detection using SAM and Faster R-CNN to separate known from novel objects, (2) Embedding–Label Repository (ELR) construction via context-aware characterization, LLM-based candidate label generation, and CLIP-driven cross-modal re-ranking, and (3) iterative refinement of assignments through neighborhood voting and visual cohesion analysis.

known categories evaluates each region proposal (we use Faster R-CNN in our experiments, though any detector can be employed) and produces a confidence score $c_i$ for each proposal $r_i$. We retain as known objects those proposals we confidently recognized as belonging to a known class, $c_i \geq \tau_{\text{conf}}$. By contrast, we flag as **candidate unknowns** proposals generated by SAM that the detector fails to classify confidently, e.g., $c_i < \tau_{\text{conf}}$. This design ensures exhaustive coverage via SAM while filtering through the known-category decision boundary of the detector. Moreover, let $\mathcal{O} = \{o_i\}_1^{n_{\text{det}}}$ be the set of detected objects with bounding boxes $\mathcal{B} = \{\boldsymbol{b}_i\}_1^{n_{\text{det}}}$. We categorize objects into known $\mathcal{O}_{\text{known}} \subset \mathcal{O}$ and unknown $\mathcal{O}_{\text{unknown}} \subset \mathcal{O}$ subsets.

## 3.2 GENERATING THE EMBEDDING-LABEL REPOSITORY (ELR)

The ELR serves as the foundational knowledge base for COSRA. It stores visual embeddings of unknown objects and their corresponding predicted labels generated by the LLM. We divide this process into two main components.

**Attribute-Based Characterization.** To provide rich contextual cues for label generation, we extract visual attributes (color, texture, material, etc.) that characterize each unknown object's appearance. For each unknown object $o_u \in \mathcal{O}_{\text{unknown}}$ with bounding box $\boldsymbol{b}_u$, we extract visual attributes using a vision-language model (VLM). We first crop the object region from the input image

$$\boldsymbol{I}_u = \text{Crop}(\boldsymbol{I}, \boldsymbol{b}_u). \tag{1}$$

To characterize each unknown object, we maintain attribute-specific text templates for different visual properties such as color (e.g., "a red object"), texture (e.g., "a smooth surface"), and material (e.g., "made of metal"). For each attribute category, we compute similarity scores between the cropped image and all candidate text descriptions using CLIP. We then apply threshold filtering to retain only attributes with high visual-textual alignment (similarity $\geq \tau_{\text{attr}} = 0.80$), ensuring that extracted textual attributes are consistent with the visual features of the cropped object.

The resulting attribute set $\mathbf{C}_u$ contains the filtered descriptors for each unknown object, providing rich contextual information for subsequent label generation.

We offer a full description of these extractions and a full list of text characteristics, e.g., shape, pattern, etc, in Appendix B.

**Scene Context Representation.** Let $\mathcal{O}_{\text{known}} \subset \mathcal{O}$ denote the detected objects with known labels. For each $o_i \in \mathcal{O}_{\text{known}}$, let $L_i$ be its known class label and $\boldsymbol{b}_i = (x_{\min}, y_{\min}, x_{\max}, y_{\max})$ its bounding box. We define the scene context set as

$$\mathcal{K} = \{ (L_i, \boldsymbol{b}_i) \mid o_i \in \mathcal{O}_{\text{known}} \}. \tag{2}$$

**Context-Aware Prompt Construction and LLM-based Candidate Label Generation.** We build a single structured Context-Aware prompt $\mathcal{P}$ that concatenates (i) the scene context $\mathcal{K}$, (ii) the unknown object's box $\boldsymbol{b}_u$ and attributes $\mathbf{C}_u$, and (iii) an instruction that specifies the task and output

format (ranked list of $m$ candidate labels). Formally,

$$\mathcal{P} = \textsc{AssemblePrompt}\big(\mathcal{K}, \mathbf{C}_u, \boldsymbol{b}_u, m\big). \tag{3}$$

We show a schematic example of the rendered prompt in Appendix C.

Eq. 3 creates the text we pass to the LLM, including the structured context plus the instruction block above. Passing $\mathcal{P}$ to the LLM yields a ranked list of $m$ candidate labels for the unknown object,

$$\mathcal{L}_u^{\text{LLM}} = \text{LLM}(\mathcal{P}) = \big(\boxed{l_{u,1}^{\textbf{LLM}}}, \ \boxed{l_{u,2}^{\textbf{LLM}}}, \ \ldots, \ \boxed{l_{u,m}^{\textbf{LLM}}}\big) \tag{4}$$

where ordering reflects the LLM's confidence i.e., the first candidate is the most likely label according to the LLM. Unless otherwise stated, we use $m = 50$, chosen via an ablation sweep $m \in \{10, 25, 50, 100, 150\}$. This ablation study demonstrates that $m = 50$ achieves near-optimal accuracy with a modest runtime (see Appendix F.1). This procedure creates a probability-like ranked hypothesis set, which we later refine through cross-modal re-ranking and Iterative Label Refinement, without using any ground-truth label of the unknown object at this stage.

**Cross-Modal Re-ranking.** We employ CLIP's dual-encoder architecture to re-rank LLM-generated candidates based on visual-semantic alignment. For unknown object $u$ with candidate labels $\mathcal{L}_u^{\text{LLM}}$, we first extract visual embeddings using the image encoder

$$\boldsymbol{v}_u = \text{Image Encoder}(\boldsymbol{I}_u), \tag{5}$$

and compute text embeddings for each candidate using the text encoder

$$t_{u,k} = \text{Text Encoder}(\boxed{l_{u,k}^{\textbf{LLM}}}), \quad k = 1, \ldots, m. \tag{6}$$

We then compute visual-textual alignment scores through similarity computation, following

$$s_{u,k} = \frac{\boldsymbol{v}_u^\top t_{u,k}}{||\boldsymbol{v}_u||_2 ||t_{u,k}||_2}. \tag{7}$$

Finally, we re-rank candidates in descending order by their scores, $s_{u,k}$, producing the final label list

$$\mathcal{L}_u = \text{Re-rank}(\mathcal{L}_u^{\text{LLM}}, \boldsymbol{v}_u) = \big(\boxed{l_{u,1}}, \ \boxed{l_{u,2}}, \ \ldots, \ \boxed{l_{u,m}}\big). \tag{8}$$

**Embedding-Label Repository Organization.** The ELR serves as our framework's knowledge base, storing visual embeddings and corresponding label predictions. We maintain the ELR as paired collections,

$$\boldsymbol{V} = \begin{bmatrix} \boldsymbol{v}_1 \\ \boldsymbol{v}_2 \\ \vdots \\ \boldsymbol{v}_N \end{bmatrix}, \quad \mathcal{L} = \begin{bmatrix} \mathcal{L}_1 \\ \mathcal{L}_2 \\ \vdots \\ \mathcal{L}_N \end{bmatrix} = \begin{bmatrix} \boxed{l_{1,1}}, \boxed{l_{1,2}}, \ldots, \boxed{l_{1,m}} \\ \boxed{l_{2,1}}, \boxed{l_{2,2}}, \ldots, \boxed{l_{2,m}} \\ \vdots \\ \boxed{l_{N,1}}, \boxed{l_{N,2}}, \ldots, \boxed{l_{N,m}} \end{bmatrix}, \quad \boldsymbol{V} \in \mathbb{R}^{N \times d}, \tag{9}$$

where $N$ is the total number of processed objects and $d$ is the embedding dimensionality. Each $\vec{v}_u$ is associated with its corresponding re-ranked label list $\mathcal{L}_u = (l_{u,1}, l_{u,2}, \ldots, l_{u,j}, \ldots, l_{u,m})$ where $j$ indicates the ranking position.

### 3.3 Stage 3: Iterative Label Refinement

Our refinement algorithm leverages semantic consistency principles, operating on the insight that objects with similar predicted labels should exhibit similar visual characteristics. The algorithm proceeds through three phases.

**Semantic Grouping.** We first partition objects into semantically coherent groups by leveraging CLIP's text encoder to compute similarities between predicted labels. Objects with semantically similar top-1 predictions are clustered together such that

$$G_j = \{u : \text{sim}(l_{u,1}, l_{j,1}) \geq \tau_{\text{sim}}\}. \tag{10}$$

**Outlier Detection.** Within each semantic group $G_j$, we identify potential mislabeled objects by detecting visual outliers whose embeddings deviate significantly from the group's visual coherence. We compute the group centroid in the visual embedding space with

$$\boldsymbol{\mu}_j = \frac{1}{|G_j|} \sum_{u \in G_j} \boldsymbol{v}_u. \tag{11}$$

Objects whose visual similarity to the centroid falls below the group's average similarity $\bar{s}_j$ are flagged as potential mislabels, i.e., $\text{Outlier}(u) = \text{sim}(\boldsymbol{v}_u, \boldsymbol{\mu}_j) < \bar{s}_j$. This principle captures the semantic-visual consistency assumption that objects with similar labels should occupy similar regions in the visual embedding space.

**Label Reassignment.** For each flagged outlier, we perform label correction through k-nearest neighbor search in the visual embedding space. We retrieve the k most visually similar objects from the ELR and apply majority voting among their current labels to determine the reassigned label. This visual similarity-based relabeling corrects semantic inconsistencies by leveraging the accumulated knowledge in the repository. The refinement process iterates until label assignments converge or maximum iterations are reached, ensuring stable and consistent label predictions across the repository. The pseudocode is provided in Appendix D.

## 4 Experiments

We validate COSRA on multiple benchmarks against closed-set (fixed label space), open-vocabulary (requires a provided category list during inference), and open-ended (no label list; names are generated during inference) detectors. Unlike open-vocabulary methods that rely on predefined vocabularies and extensive training, COSRA requires neither category supervision nor training, making it well-suited for open-world scenarios.

### 4.1 Experimental Setup

**Datasets.** We evaluate our framework on two widely used benchmarks, COCO (Lin et al., 2014) and LVIS (Gupta et al., 2019), which cover diverse object categories with varying frequency distributions. For COCO, we adopt the OV-RCNN (Zareian et al., 2020) protocol, splitting the categories into 48 base and 17 novel classes. For LVIS, we follow the ViLD (Gu et al., 2021) split, partitioning 1,203 categories by annotation frequency: 337 rare classes serve as novel, while the remaining 866 common and frequent classes form the base set. This split reflects real-world long-tail distributions where many objects appear infrequently. To enable comparison with open-ended detection models, we also evaluate on LVIS minival, which is the 5k COCO val2017 images re-annotated with the full LVIS label set.

**Evaluation Protocol.**

Our evaluation follows a two-phase protocol to ensure fair assessment of open-world detection. In the repository construction phase, we build the Embedding–Label Repository (ELR) by processing both training and validation sets, collecting objects that the pretrained detector fails to recognize with confidence below threshold $\tau_{\text{conf}}$. This guarantees that the ELR contains only unknown objects, avoiding contamination from known categories. We apply this procedure to COCO and LVIS, yielding dataset-specific repositories with 226,542 unknown objects for COCO and 87,293 for LVIS. In the evaluation phase, we test COSRA on validation objects while keeping repository construction and evaluation strictly separated. This setup ensures that reported performance reflects true generalization to unseen validation instances, while still benefiting from the broad coverage of unknowns in the ELR. We compute standard detection metrics by matching COSRA's predicted unknown objects against ground-truth annotations. For datasets with predefined vocabularies, we follow GenerateU and VL-SAM by mapping COSRA's generated labels to canonical categories using CLIP text–image similarity with the template "a object category", enabling fair comparison with existing benchmarks while retaining open-ended label generation.

**Implementation Details and Evaluation Metrics.** We use Faster R-CNN with ResNet50-C4 as the detection backbone. For COCO experiments, we use a model pretrained on 48 base categories following OV-RCNN (Zareian et al., 2020). For LVIS experiments, we use a model pretrained on 866 common and frequent classes, excluding the 337 rare categories. We set a confidence threshold of $\tau_{\text{conf}} = 0.9$ for known object detection, following established practice in semi-supervised object detection where this threshold ensures high precision for pseudo label filtering (Liu et al., 2023b). SAM provides region proposals with $32\times32$ grid sampling. For label generation, we use LLaMA-4 (Touvron et al., 2023) ($\text{temperature} = 0.35$) generating $m = 50$ candidates, and CLIP ViT-B/32 for cross-modal re-ranking ($\tau_{\text{attr}} = 0.80$). The iterative refinement uses semantic grouping ($\tau_{\text{sim}} = 0.85$) with $k = 30$ neighbors and 12/17 iterations for COCO/LVIS. We detail the complete

| Method | Type | Training-free | $\text{AP}_{rare}$ (%) |
|---|---|---|---|
| Mask R-CNN (He et al., 2017) | Close-Set | × | 26.3 |
| Deformable DETR (Zhu et al., 2020) | | × | 24.2 |
| GLIP (Li et al., 2022b) | Open-Vocabulary | × | 20.8 |
| GroundingDINO (Liu et al., 2024) | | × | 27.4 |
| DetCLIP (Yao et al., 2022) | | × | 26.9 |
| YOLOWorld (Cheng et al., 2024) | | × | 27.1 |
| **COSRA**Open-Vocabulary | | ✓ | **32.9** |
| GenerateU (Lin et al., 2024a) | Open-Ended | × | 20.0 |
| Open-Det (Cao et al., 2025) | | × | 21.9 |
| VL-SAM (Lin et al., 2024b) | | ✓ | 23.4 |
| **COSRA (Ours)** | | ✓ | **27.1** |

**Table 1: Comparison of object detection results on LVIS minival**. "Open-Ended" indicates inference without predefined object categories, while "Open-vocabulary" uses predefined category lists to constrain generation.

| Method | Training-free | Vocabulary-free | Novel/Rare (%) | Overall (%) |
|---|---|---|---|---|
| **COCO Dataset (AP)** | | | | |
| ViLD (Gu et al., 2021) | × | × | 27.6 | 51.3 |
| Detic (Zhou et al., 2022) | × | × | 27.8 | 44.9 |
| RegionCLIP (Zhong et al., 2022) | × | × | 26.8 | 47.5 |
| VLDet (Lin et al., 2023) | × | × | 32.0 | 45.8 |
| BARON (Wu et al., 2023) | × | × | 33.1 | 49.1 |
| SAMP (Zhao et al., 2024) | × | × | 34.8 | 54.2 |
| DVDet (Jin et al., 2024) | × | × | 35.8 | 57.0 |
| **COSRA (Ours)** | ✓ | ✓ | **38.8** | **60.2** |
| **LVIS v1 Dataset (mAP mask)** | | | | |
| ViLD (Gu et al., 2021) | × | × | 16.6 | 25.5 |
| DetPro (Du et al., 2022) | × | × | 19.8 | 25.9 |
| RegionCLIP (Zhong et al., 2022) | × | × | 17.1 | 28.2 |
| VLDet (Lin et al., 2023) | × | × | 21.7 | 30.1 |
| BARON (Wu et al., 2023) | × | × | 19.2 | 26.5 |
| DVDet (Jin et al., 2024) | × | × | 23.1 | 31.2 |
| **COSRA (Ours)** | ✓ | ✓ | **24.2** | **33.7** |

**Table 2: Comparison of object detection results on COCO and LVIS**. All methods use ResNet50 as the backbone. "Novel" refers to unseen categories in COCO, while "Rare" refers to low-frequency categories in LVIS. COSRA achieves the best performance without requiring training or predefined vocabularies.

hyperparameter settings in Appendix E. We report mAP at IoU 0.5 (Zareian et al., 2021) for COCO, mask AP (Lin et al., 2023) for LVIS, and box AP at IoU 0.5 (Zareian et al., 2021) for LVIS minival.

## 4.2 MAIN RESULTS

**Evaluation on LVIS Minival.** In the open-ended setting, COSRA achieves 27.1 $\text{AP}_{rare}$ on LVIS minival, surpassing all open-ended approaches including VL-SAM, Open-Det, and GenerateU. We also adapt COSRA to the Open-vocabulary scenario by providing the list of LVIS categories in the LLM prompt, resulting in COSRAOpen-Vocabulary with 32.9 $\text{AP}_{rare}$ as shown in Table 1. Notably, COSRA outperforms all Open-vocabulary models. Among all compared methods, VL-SAM represents the most comparable baseline due to being training-free, yet it relies directly on a vision-language model (VLM) to enumerate possible object names in free-form text, making its predictions highly dependent on the VLM's immediate outputs and prone to inconsistency. In contrast, COSRA adopts a structured, context-aware prompting strategy where the LLM proposes candidate labels for unknown regions that are subsequently refined through cross-modal similarity and iterative self-refinement, resulting in superior performance across challenging long-tail distributions.

**Evaluation on COCO.** COSRA achieves remarkable results with 38.8 Novel AP (Table 2). Compared to the strongest baseline DVDet (35.8 Novel AP), COSRA provides 8.4% relative improvement in Novel AP. These gains are particularly significant considering COSRA's training-free nature, while DVDet requires extensive training on large-scale datasets. Against other competitive methods, COSRA substantially outperforms BARON (33.1 Novel AP) by 17.2% and SAMP (34.8 Novel AP) by 11.5%, establishing clear superiority in novel object recognition.

**Evaluation on LVIS** On the more challenging LVIS dataset with its long-tail distribution, COSRA achieves 24.2 Rare AP (Table 2). Compared to DVDet (23.1 Rare AP), COSRA provides 4.8%

improvement in Rare AP. Against other strong baselines, COSRA outperforms VLDet (21.7 Rare AP) by 11.5% and DetPro (19.8 Rare AP) by 22.2%. These results demonstrate COSRA's robustness across diverse object categories and frequency distributions.

**Qualitative Results.** Fig. 2 demonstrates COSRA's iterative refinement capabilities across multiple unknown objects in a single scene. The primary objective of open-ended detection is to assign the most accurate semantic label to each provided bounding box containing unknown objects. In this kitchen scene, we observe several key refinement behaviors: initially, due to partial occlusion, the sink is mislabeled as `sponge` based on VLM and LLM outputs (since the visible sponge represents a prominent part of the occluded sink). However, COSRA corrects this to `sink` in the first iteration. Notably, three objects are successfully corrected within the first iteration, while the `pipe` requires additional contextual evidence and is refined to `tap` only in iteration 12. This demonstrates COSRA's ability to handle multiple object corrections simultaneously while effectively managing noise arising from occlusion and partial visibility challenges inherent in real-world scenarios. Additional refinement examples demonstrating various challenging scenarios are provided in Appendix H, with detailed noise handling analysis presented in Appx. G.

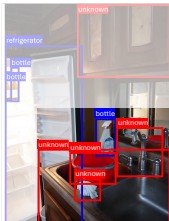 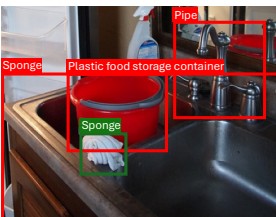 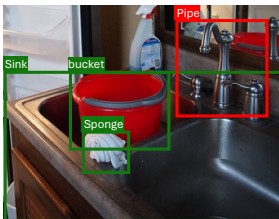 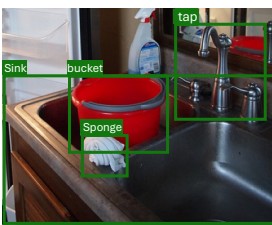

| Initial Detection | Label Assignment (Before Refinement) | Iterative Refinement (iter 1) | Iterative Refinement (iter 12) |

**Figure 2: COSRA's labeling process across multiple unknown objects.** Blue labels show known objects detected by the detector, red labels indicate objects requiring further processing (unknown objects initially, then incorrect/partially correct predictions), and green labels show objects with correct final predictions. While initially predicting `sponge` correctly, during self-refinement stages it correctly relabeled `sink` and `bucket` (iteration 1) and `tap` (iteration 12), demonstrating robustness to initial prediction errors from occlusion and partial visibility, with multi-object correction capabilities in single iterations.

## 4.3 ADDITIONAL ANALYSIS

**Sensitivity to the ELR's Size.** We systematically examine how the size of the Embedding–Label Repository affects detection performance across varying repository scales. Starting from 5% of the full repository and scaling progressively, our analysis reveals that both COCO and LVIS datasets exhibit logarithmic improvement patterns as repository size increases (Figure 3 left). This behavior validates our hypothesis that expanding the repository enhances the representational diversity of visual embeddings while simultaneously reducing semantic ambiguity in label predictions. The observed scaling properties highlight the critical role of repository comprehensiveness in enabling effective open-world object detection, particularly for handling the distributional complexity inherent in real-world visual scenarios.

**Effect of Contextual Information on the performance.** The contextual reasoning analysis (Figure 3 right) reveals the fundamental importance of scene context in novel object detection. These results, obtained before applying iterative refinement, demonstrate the direct effects of contextual information on label prediction accuracy. Scenes with no known objects achieve only 6% accuracy for COCO and 4% for LVIS, as the method relies solely on visual characteristics. Performance improves dramatically with contextual cues: 1-2 objects (19% COCO, 11% LVIS), 3-4 objects (23% COCO, 14% LVIS), 5-7 objects (27% COCO, 17% LVIS), and peaks at 8+ objects (29% COCO, 18% LVIS). This demonstrates contextual information as a fundamental component for accurate novel object detection, with improvements of up to 23 percentage points for COCO and 14 percentage points for LVIS when sufficient contextual cues are available.

## 4.4 ABLATION STUDY

We validate the contribution of each COSRA component with an ablation study. Scene context provides the most critical contribution, confirming our entropy-based motivation. We found all components—iterative refinement, attribute characterization, and cross-modal re-ranking—contribute

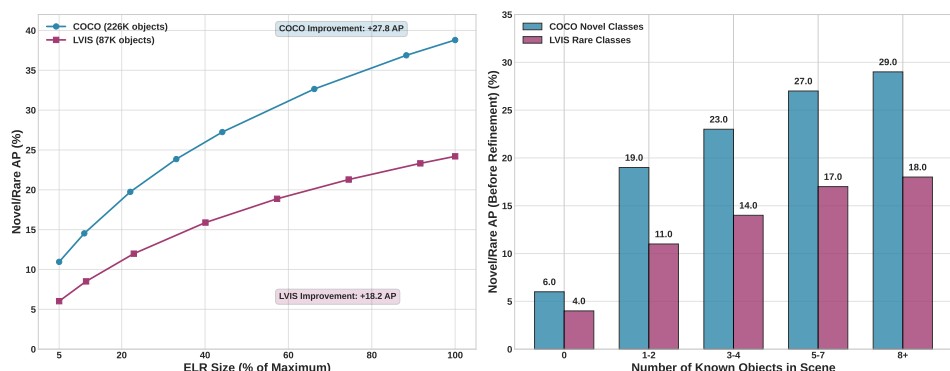

**Figure 3: Effect of repository size and scene context on object detection.** (Left) Larger ELR steadily improves performance with +27.8 pp on COCO and +18.2 pp on LVIS. (Right) Detection accuracy improves with more known objects in the scene, revealing the importance of contextual cues for open-ended labeling.

significantly to overall performance. We summarize these results in Table 3 and detail them in Appendix F.3.

| Method | COCO Novel AP (%) | LVIS Rare AP (%) |
|---|---|---|
| w/o Scene Context | 7.1 | 4.2 |
| w/o Attribute Characterization | 23.1 | 15.2 |
| w/o Cross-Modal Re-ranking | 27.1 | 16.0 |
| w/o Iterative Refinement | 23.0 | 14.1 |
| COSRA (Full) | **38.8** | **24.2** |

**Table 3: Ablation study on COSRA components**. Each row shows the effect of removing a specific component from the full framework.

### 4.5 CONCLUSIONS AND LIMITATIONS

**Limitations.** COSRA has two main areas for improvement. First, our fixed attribute templates may limit the richness of object characterization in certain domains. This could be enhanced through dynamic attribute generation, where LLMs create environment-specific descriptors tailored to the contexts of detected objects. Second, while leveraging foundation models enables our training-free approach, COSRA's performance is naturally tied to the capabilities of these underlying models. Future work could explore ensemble approaches, such as combining SAM with complementary region proposal methods (e.g., EdgeBoxes (Zitnick & Dollár, 2014), MCG (Arbeláez et al., 2014)) or integrating multiple vision-language encoders (e.g., ALIGN (Jia et al., 2021), BLIP-2 (Li et al., 2023)) to achieve more robust multi-modal representations and enhanced performance.

**Conclusions.** We present COSRA, a training-free framework that addresses the fundamental challenge of open-ended object detection through context-aware reasoning and self-refining mechanisms. By leveraging entropy-driven contextual analysis, COSRA assigns semantic labels to previously unseen objects without requiring predefined vocabularies or supervised training on novel categories. Our approach demonstrates that contextual information can effectively constrain the semantic space for unknown objects, enabling accurate label generation through iterative refinement processes that progressively improve predictions via visual similarity clustering and neighborhood consensus. Extensive evaluation validates COSRA's effectiveness, substantially outperforming existing approaches across multiple benchmarks while maintaining complete vocabulary independence. These results establish a new paradigm for autonomous object discovery in open-world scenarios, demonstrating that training-free, context-driven reasoning can rival supervised methods in novel object recognition tasks. COSRA's success suggests promising directions for scalable, adaptive computer vision systems that can continuously expand their understanding without explicit retraining.

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

# A    THEORETICAL MOTIVATION

A core theoretical principle behind COSRA is grounded in information theory, particularly the use of conditional entropy to model contextual uncertainty. Let $Y$ denote the unknown label of an object and $X_1, X_2, \ldots, X_n$ represent the observed contextual variables (e.g., known object labels, spatial locations, or visual attributes) in the scene. The uncertainty in predicting $Y$ given these variables is captured by the conditional entropy:

$$H(Y \mid X_1, X_2, \ldots, X_n)$$

From Shannon's Inequality, we know that as more informative context is added, this entropy cannot increase (Mitrinović et al., 1993):

$$H(Y \mid X_1, X_2, \ldots, X_n) \leq H(Y \mid X_1, X_2, \ldots, X_{n-1})$$

This submodular property of entropy implies that the inclusion of each new contextual object narrows the plausible space of labels for $Y$. In COSRA, this principle is operationalized through prompt engineering: every new known object added to the prompt provides semantic and spatial constraints that help the LLM disambiguate the label for the unknown object.

# B    CHARACTERISTIC EXTRACTION USING CLIP

This appendix provides comprehensive details of our attribute-based characterization process, a critical component of the Embedding-Label Repository (ELR) construction pipeline. Our approach systematically extracts visual and semantic descriptors for unknown objects using CLIP's multimodal capabilities.

## B.1    TEMPLATE-BASED EXTRACTION FRAMEWORK

Our characteristic extraction operates on a template-based framework where each attribute category $\chi$ maintains a curated set of descriptors $\mathcal{T}_\chi$. For an unknown object $o_u$ with cropped region $\boldsymbol{I}_u = \mathrm{Crop}(\boldsymbol{I}, \boldsymbol{b}_u)$, we evaluate each descriptor $t \in \mathcal{T}_\chi$ by computing the cosine similarity:

$$S_\chi(t) = \frac{\langle f_{\mathrm{img}}(\boldsymbol{I}_u), f_{\mathrm{text}}(t) \rangle}{||f_{\mathrm{img}}(\boldsymbol{I}_u)||_2 \cdot ||f_{\mathrm{text}}(t)||_2} \tag{12}$$

where $f_{\mathrm{img}}$ and $f_{\mathrm{text}}$ denote CLIP's image and text encoders, respectively.

We retain descriptors exceeding threshold $\tau_{\mathrm{attr}} = 0.80$ to ensure confident attribute assignments that meaningfully contribute to contextual reasoning. Our framework encompasses 11 distinct characteristic categories with 155 total descriptors, systematically covering visual appearance, functional properties, and contextual attributes.

## B.2    CHARACTERISTIC CATEGORIES

We organize our characteristic categories into three primary groups based on their semantic function. Tables 4–6 provide comprehensive specifications for each category.

**Visual Properties.**    Visual characteristics capture observable appearance attributes that directly correspond to object recognition cues.

**Functional Properties.**    Functional characteristics describe operational capabilities and material composition that inform object utility.

**Table 4: Visual property characteristics and their descriptors.**

| Category | Template | Descriptors |
|---|---|---|
| Color (19) | "{color} colored" | Red, Blue, Green, Yellow, Orange, Purple, Pink, Brown, Black, White, Gray, Cyan, Magenta, Teal, Lavender, Silver, Chrome, Neon, Multi-colored |
| Shape (18) | "in {shape} shape" | Circle, Square, Triangle, Rectangle, Oval, Cylindrical, Aerodynamic, Spherical, Octagonal, Conical, Star, Heart, Crescent, Diamond, Trapezoid, Parallelogram, Rhombus, Unknown |
| Texture (12) | "with {texture} texture" | Smooth, Rough, Bumpy, Fuzzy, Shiny, Matte, Textured, Polished, Grainy, Glossy, Weathered, Rusted |
| Pattern (8) | "with {pattern} pattern" | Solid, Striped, Spotted, Checkered, Camouflage, Plain, Branded, Reflective |
| Size (6) | "a {size} sized object" | Tiny, Small, Medium, Large, Huge, Massive |

**Table 5: Functional property characteristics and their descriptors.**

| Category | Template | Descriptors |
|---|---|---|
| Use Cases (24) | "used for {use_case}" | Rest, Work, Entertainment, Storage, Cooking, Cooling, Eating, Cleaning, Decoration, Transportation (Public/Private), Communication, Exercise, Studying, Personal Care, Safety, Signaling, Warning, Parking, Sporting, Racing, Emergency, Commercial, Passenger, Cargo |
| Materials (20) | "made of {material} material" | Metal, Steel, Aluminum, Plastic, Rubber, Glass, Leather, Concrete, Fiberglass, Carbon Fiber, Wood, Fabric, Fur, Feathers, Ceramic, Paper, Cardboard, Foam, Silicone, Unknown |
| Functionality (14) | "used for {functionality} purpose" | Foldable, Adjustable, Portable, Stationary, Electronic, Mechanical, Multi-functional, Decorative, Mobile, Automated, Manual, Emergency, Recreational, Commercial |

**Contextual Properties.** Contextual characteristics establish environmental and categorical frameworks that constrain semantic possibilities.

**Table 6: Contextual property characteristics and their descriptors.**

| Category | Template | Descriptors |
|---|---|---|
| Categories (18) | "classified as {category}" | Vehicle, Automotive Vehicle, Living Being, Furniture, Electronic Device, Sports Equipment, Kitchen Appliance, Tool, Clothing, Building Structure, Natural Object, Artwork, Container, Weapon, Musical Instrument, Medical Equipment, Toy, Book, Food Item |
| Environment (9) | "found in {environment} environment" | Urban, Rural, Marine, Aerial, Terrestrial, All-terrain, Road, Rail, Water |
| Age/Era (6) | "from {age} era" | Modern, Vintage, Antique, Contemporary, Retro, Weathered |

## B.3 INTEGRATION WITH COSRA FRAMEWORK

The extracted characteristics $\mathbf{C}_u = \{\mathbf{C}_{u,\text{color}}, \mathbf{C}_{u,\text{shape}}, \ldots, \mathbf{C}_{u,\text{category}}\}$ serve as essential contextual variables in our entropy-driven framework. As presented in Appendix A, each characteristic category contributes to reducing the conditional entropy $H(Y \mid X_1, \ldots, X_n)$ of label prediction, where $Y$ represents the semantic label and $X_i$ denotes individual characteristic variables.

During context-aware prompt construction (Equation 3), selected characteristics are systematically integrated with scene context $\mathcal{K}$ and spatial information $b_u$. This multi-modal integration enables the LLM to leverage both visual attributes and contextual relationships when generating candidate labels $\mathcal{L}_u^{\text{LLM}}$.

The comprehensive characteristic framework ensures semantic consistency between CLIP-based visual similarity assessments and LLM-generated textual hypotheses. Visual properties directly inform appearance-based similarities, functional properties constrain usage-based reasoning, and contextual properties establish environmental and categorical priors that guide semantic inference. This systematic organization enables COSRA to effectively handle diverse object types across various domains while maintaining computational efficiency through threshold-based filtering ($\tau_{\text{attr}} = 0.80$) that preserves only the most confident attribute assignments.

## C  PROMPT EXAMPLE

To propose labels for novel regions, we prompt a general-purpose LLM with scene context from detected known objects and the unknown region's bounding box and attributes. The instruction below asks for *m* candidate names ranked by plausibility. The resulting candidates are re-ranked with CLIP and stored in the Embedding–Label Repository (ELR).

---

**Candidate Label Generation Prompt**

**Known objects (label, box):**
$(L_1, b_1), (L_2, b_2), \ldots, (L_{|\mathcal{K}|}, b_{|\mathcal{K}|})$
**Unknown object:**
box $= b_u$
attributes $= \mathbf{C}_u$ *(e.g., "red, glossy, metallic, striped")*
**Task (instruction):**
Based on the scene context and the described characteristics of the unknown object, provide $\{m\}$ possible names for the object, ranked from most likely to least likely. Return only the list of names, one per line.

---

## D  LABEL REFINEMENT PROCESS

This section provides the complete algorithmic details for the iterative label refinement process described in Section 3.3. The refinement algorithm leverages semantic consistency principles to progressively correct noisy label assignments through visual similarity clustering and neighborhood voting.

The algorithm operates on the insight that objects with similar predicted labels should exhibit similar visual characteristics in the embedding space. It proceeds through three main phases: (1) semantic grouping based on label similarity, (2) outlier detection within each group, and (3) label reassignment via k-nearest neighbor voting. The process iterates until convergence or a maximum number of iterations is reached.

**Algorithm 1** Label Refinement process

**Require:** Visual embeddings $\{V_u\}_{u=1}^N$, $V_u \in \mathbb{R}^{512}$
**Require:** Initial predicted labels $\{L_u^0\}$ for unknown objects
**Require:** Hyperparameters: $\tau_{\text{sem}}=0.85$, $k_{\text{neighbors}}=30$, $T=3$
**Ensure:** Refined labels $\{L_u^*\}$
1: $t \leftarrow 0$; $\quad converged \leftarrow$ false
2: $L_u^t \leftarrow L_u^0 \quad \forall u$
3: **while** $t < T$ **and** $converged =$ false **do**
4: $\quad t \leftarrow t + 1$
5: $\quad changes \leftarrow 0$
6: $\qquad\qquad\qquad\qquad\qquad\qquad\qquad\qquad\qquad$ ▷ **STAGE 1: CLIP-Based Semantic Grouping**
7: $\quad \mathcal{G} \leftarrow \text{GroupByCLIPSimilarity}\big(\{L_u^{t-1}\}, \tau_{\text{sem}}\big)$
8: $\qquad\qquad\qquad\qquad\qquad$ ▷ **STAGE 2: Group-Average-Based Flagging and Relabeling**
9: $\quad \mathcal{F} \leftarrow \emptyset$ $\qquad\qquad\qquad\qquad\qquad\qquad\qquad\qquad\qquad\qquad$ ▷ Flagged objects
10: $\quad$ **for** each semantic group $g \in \mathcal{G}$ **do**
11: $\qquad$ **if** $|g| < 2$ **then** $\qquad\qquad\qquad\qquad\qquad\qquad\qquad$ ▷ Skip singleton groups
12: $\qquad\quad$ **continue**
13: $\qquad$ **end if**
14: $\qquad \mathcal{E}_g \leftarrow \{V_u \mid u \in g\}$ $\qquad\qquad\qquad\qquad\qquad\qquad$ ▷ Embeddings in group
15: $\qquad S \leftarrow \text{CosineSimilarityMatrix}(\mathcal{E}_g)$
16: $\qquad$ **for** each $u \in g$ **do**
17: $\qquad\quad \text{avg\_sim}_u \leftarrow \frac{1}{|g|-1} \sum_{v \in g, v \neq u} S_{u,v}$
18: $\qquad$ **end for**
19: $\qquad \text{group\_avg} \leftarrow \frac{1}{|g|} \sum_{u \in g} \text{avg\_sim}_u$
20: $\qquad$ **for** each $u \in g$ **do**
21: $\qquad\quad$ **if** $\text{avg\_sim}_u < \text{group\_avg}$ **then**
22: $\qquad\qquad \mathcal{F} \leftarrow \mathcal{F} \cup \{u\}$ $\qquad\qquad\qquad\qquad\qquad$ ▷ Flag for relabeling
23: $\qquad\quad$ **end if**
24: $\qquad$ **end for**
25: $\quad$ **end for**
26: $\qquad\qquad\qquad\qquad\qquad\qquad\qquad\qquad$ ▷ **k-NN Relabeling for Flagged Objects**
27: $\quad$ **for** each flagged object $u \in \mathcal{F}$ **do**
28: $\qquad \mathcal{N}_u \leftarrow \text{FindKNN}(V_u, k_{\text{neighbors}})$ $\qquad\qquad\qquad\qquad$ ▷ FAISS-based kNN
29: $\qquad \text{labels} \leftarrow \{L_v^{t-1} \mid v \in \mathcal{N}_u, v \neq u\}$
30: $\qquad L_u^{\text{new}} \leftarrow \text{MajorityVote}(\text{labels})$
31: $\qquad$ **if** $L_u^{\text{new}} \neq L_u^{t-1}$ **then**
32: $\qquad\quad L_u^t \leftarrow L_u^{\text{new}}; \quad changes \leftarrow changes + 1$
33: $\qquad$ **else**
34: $\qquad\quad L_u^t \leftarrow L_u^{t-1}$
35: $\qquad$ **end if**
36: $\quad$ **end for**
37: $\qquad\qquad\qquad\qquad\qquad\qquad$ ▷ **Keep non-flagged objects unchanged**
38: $\quad$ **for all** $u \notin \mathcal{F}$ **do**
39: $\qquad L_u^t \leftarrow L_u^{t-1}$
40: $\quad$ **end for**
41: $\qquad\qquad\qquad\qquad\qquad\qquad\qquad\qquad\qquad\qquad$ ▷ **Convergence check**
42: $\quad$ **if** $changes = 0$ **then**
43: $\qquad converged \leftarrow$ true
44: $\quad$ **end if**
45: **end while**
46: **return** $\{L_u^*\} \leftarrow \{L_u^t\}$

# E  IMPLEMENTATION DETAILS AND HYPERPARAMETERS

## E.1  EXPERIMENTAL SETUP

This section provides additional implementation details beyond those presented in the main text. For region proposals, SAM uses the specific threshold settings shown in Table 7. These parameters were chosen to balance proposal quality with computational efficiency, ensuring comprehensive coverage of salient regions while maintaining reasonable inference times.

Table 7: SAM hyperparameter settings used for region proposals.

| Parameter | Value |
|---|---|
| Prediction IoU threshold | 0.88 |
| Stability score threshold | 0.95 |
| NMS threshold | 0.70 |

## E.2  COSRA HYPERPARAMETER SETTINGS

For the VLM component, we use CLIP ViT-B/32 as the backbone for both visual and textual processing, applying CLIP's image encoder $f_{img}$ to cropped object regions and its text encoder $f_{text}$ to attribute templates and candidate labels during cross-modal re-ranking, with a similarity threshold of $\tau_{attr} = 0.80$ for attribute selection. Table 8 summarizes the complete hyperparameter settings across all framework components, including attribute selection, semantic grouping, label generation, and iterative refinement stages. Values were chosen based on comprehensive ablation studies (Section F) for $\tau_{sim}$, $m$, and $k$, and convergence analysis for iteration counts.

Table 8: Key hyperparameter settings for COSRA framework components.

| Parameter | Value |
|---|---|
| Attribute selection threshold ($\tau_{attr}$) | 0.80 |
| Semantic grouping threshold ($\tau_{sim}$) | 0.85 |
| Number of candidate labels ($m$) | 50 |
| k-NN neighbors ($k$) | 30 |
| Refinement iterations (COCO / LVIS) | 12 / 17 |

# F  ABLATION STUDIES

## F.1  CANDIDATE LABEL COUNT ANALYSIS

We conduct a systematic analysis to determine the optimal number of candidate labels $m$ generated by the LLM during the label generation phase. Table 9 presents performance across different values of $m \in \{10, 25, 50, 100, 150\}$ on the COCO dataset.

Table 9: Effect of candidate label count $m$ on COCO detection performance. Results show diminishing returns beyond $m = 50$ with increased computational overhead.

| Candidate Count ($m$) | COCO Novel AP (%) | Runtime (s/image) |
|---|---|---|
| 10 | 35.2 | 2.1 |
| 25 | 37.4 | 2.8 |
| 50 | **38.8** | 3.4 |
| 100 | 38.9 | 5.7 |
| 150 | 39.0 | 8.2 |

The analysis reveals that $m = 50$ achieves near-optimal performance while maintaining computational efficiency. Increasing beyond $m = 50$ yields marginal improvements (+0.1-0.2 AP) at significant computational cost (67% increased runtime for $m = 100$, 141% for $m = 150$). With $m = 10$, performance drops substantially (-3.6 AP), indicating insufficient candidate diversity. The choice of $m = 25$ provides reasonable performance but still underperforms the optimal setting by -1.4 AP.

These results demonstrate that $m = 50$ strikes an optimal balance between label quality and computational efficiency, providing sufficient candidate diversity for accurate cross-modal re-ranking while avoiding the diminishing returns observed with larger candidate sets. The modest runtime increase compared to smaller values is justified by the substantial performance gains, making $m = 50$ the recommended configuration for practical deployment.

## F.2 ITERATIVE REFINEMENT CONVERGENCE ANALYSIS.

Figure 4 demonstrates the convergence behavior of our iterative refinement process across COCO and LVIS1. The COCO dataset exhibits rapid convergence, achieving stable performance after 12 iterations with a substantial improvement from 23.0% to 38.8% Novel AP. In contrast, the LVIS dataset requires additional refinement cycles, converging at iteration 17 with an improvement from 14.0% to 24.2% Novel AP. This difference in convergence behavior reflects the inherent challenge of rare class identification in LVIS, where the refinement process requires more iterations to effectively distinguish and relabel less frequent object categories.

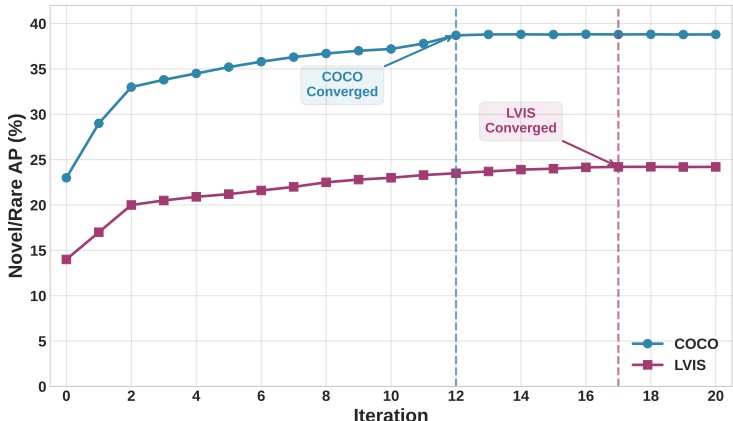

Figure 4: **Convergence behavior** of COSRA's iterative refinement on COCO and LVIS datasets. Novel/Rare AP steadily improves with each iteration, stabilizing after approximately 12 iterations on COCO and 17 iterations on LVIS. The earlier convergence on COCO reflects the relative simplicity of its class distribution, while LVIS requires more refinement steps due to its long-tail nature.

## F.3 COMPONENT ABLATION ANALYSIS

We conduct systematic ablation experiments to analyze the contribution of each component in the COSRA framework. The results in Table 3 (main paper) show the impact of removing individual components from the full system, evaluated on both COCO and LVIS datasets. Below we provide detailed analysis of each component's contribution:

**Scene Context.** Removing scene context results in the most dramatic performance drop, with COCO Novel AP falling from 38.8% to 7.1% (-31.7 pp) and LVIS Rare AP from 24.2% to 4.2% (-20.0 pp). This confirms our entropy-based motivation that contextual information is crucial for reducing label uncertainty in open-world scenarios.

**Attribute Characterization.** Without visual attribute extraction, performance decreases to 23.1% COCO Novel AP (-15.7 pp) and 15.2% LVIS Rare AP (-9.0 pp). This demonstrates that CLIP-based visual descriptors significantly enhance the LLM's ability to generate contextually appropriate candidate labels.

**Cross-Modal Re-ranking.** Removing CLIP-based re-ranking of LLM candidates reduces performance to 27.1% COCO Novel AP (-11.7 pp) and 16.0% LVIS Rare AP (-8.2 pp). This validates the importance of visual-semantic alignment in candidate selection.

**Iterative Refinement.** Without the refinement process, performance drops to 23.0% COCO Novel AP (-15.8 pp) and 14.1% LVIS Rare AP (-10.1 pp). This highlights the value of progressive label correction through visual similarity clustering and neighborhood voting.

The ablation results confirm that all components contribute meaningfully to COSRA's performance, with scene context being the most critical component, followed by iterative refinement and attribute characterization.

### F.4 HYPERPARAMETER SENSITIVITY ANALYSIS

We analyze the sensitivity of key hyperparameters in the iterative refinement process on the COCO dataset. Figure 5 presents the effect of varying the number of nearest neighbors $k$ and the label similarity threshold $\tau_{\text{sim}}$ on detection performance.

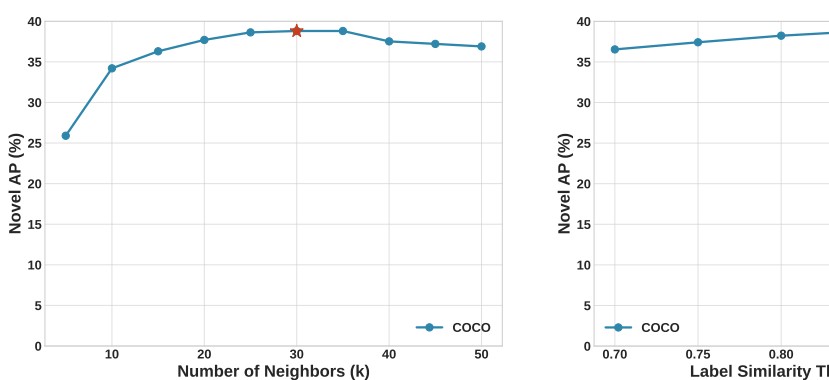

**(a)** Effect of k-neighbors on Novel AP       **(b)** Effect of $\tau_{\text{sim}}$ on Novel AP

**Figure 5: Hyperparameter sensitivity analysis.** (a) Performance varies significantly with the number of nearest neighbors $k$, peaking at $k = 30$. (b) Label similarity threshold $\tau_{\text{sim}}$ shows optimal performance at $\tau_{\text{sim}} = 0.85$.

**Number of Nearest Neighbors ($k$).** The analysis reveals that performance is highly sensitive to the choice of $k$ in the k-NN refinement process. Starting from $k = 5$ (26.0% Novel AP), performance increases substantially as we include more neighbors, reaching a peak at $k = 30$ (38.8% Novel AP). This significant improvement (+12.8 pp) demonstrates that sufficient neighborhood diversity is crucial for accurate majority voting. Beyond $k = 30$, performance gradually decreases, with $k = 50$ achieving 36.5% Novel AP (-2.3 pp from peak). This decline suggests that including too many distant neighbors introduces noise into the voting process, diluting the quality of label corrections.

**Label Similarity Threshold ($\tau_{\text{sim}}$).** The threshold analysis shows that COSRA is relatively robust to variations in the semantic grouping threshold. Performance remains stable across the tested range, with $\tau_{\text{sim}} = 0.85$ achieving optimal results (38.8% Novel AP). Lower thresholds ($\tau_{\text{sim}} = 0.7$: 36.5% Novel AP) create overly broad semantic groups, mixing dissimilar objects and reducing refinement effectiveness. Higher thresholds ($\tau_{\text{sim}} = 0.95$: 36.8% Novel AP) create overly restrictive groups, limiting the refinement process by excluding semantically related but not identical labels.

These results confirm that our chosen values of $k = 30$ and $\tau_{\text{sim}} = 0.85$ achieve peak performance, providing the optimal balance between refinement effectiveness and computational efficiency.

### G COSRA'S ROBUSTNESS TO VISUAL NOISE.

COSRA demonstrates robust performance under challenging conditions including occlusion and visual ambiguity. In open-ended object detection, the primary objective is to assign the most semantically accurate class label to each detected bounding box, rather than generating any plausible name that may be partially correct. Figure 6 provides a compelling illustration of this resilience through a complex indoor scene where partial occlusion leads to an initially incorrect prediction that is subsequently corrected through our self-refinement mechanism.

In this example, the target object (a couch) is partially occluded by a bicycle within the bounding box. The initial characteristic extraction using CLIP focuses predominantly on the visible bicycle components, leading to erroneous attribute characterization: **textures:** Grainy, **sizes:** Medium, **patterns:** Striped, **ages:** Contemporary, **functionalities:** Recreational, **materials:** Carbon Fiber, **colors:** Black, **shapes:** Aerodynamic, **use_cases:** Transportation, **environments:** Urban, **categories:**

Vehicle. This bicycle-centric characterization misleads the LLM during candidate generation, resulting in the initial incorrect prediction of `motorcycle`.

However, COSRA's iterative refinement process effectively addresses this challenge. In the first self-refinement iteration, the visual embedding of the occluded couch—despite the noise from the bicycle—maintains sufficient semantic similarity to other couch instances in the Embedding-Label Repository (ELR). Through k-nearest neighbor analysis and majority voting among visually similar objects, the refinement mechanism identifies the semantic inconsistency between the predicted `motorcycle` label and the actual visual content. The algorithm successfully relabels the object as `couch`, demonstrating COSRA's ability to overcome characteristic extraction errors and converge to semantically accurate labels even under significant visual occlusion.

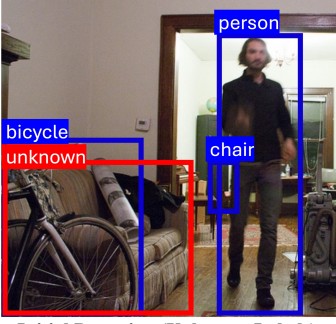 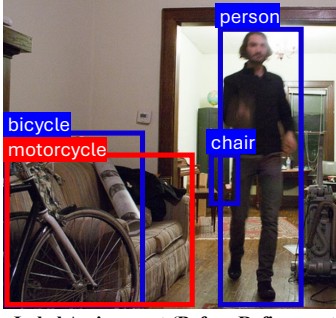 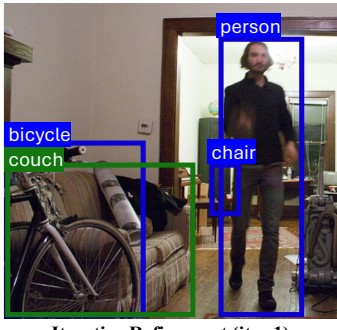

**Initial Detection (Unknown Labels)**     **Label Assignment (Before Refinement)**     **Iterative Refinement (iter 1)**

**Figure 6: COSRA's robustness to occlusion and visual noise.** `motorcycle` corrected to `couch` despite bicycle occlusion (Iter 1).

## H  ADDITIONAL QUALITATIVE EXAMPLES

### H.1  IMMEDIATE LABEL GENERATION EXAMPLES

Figure 7 demonstrates COSRA's effectiveness in immediate label generation, where our framework correctly identifies object categories even before the iterative refinement process. This showcases the strength of our entropy-driven contextual reasoning and LLM-based candidate generation. The figure illustrates two key scenarios:

In the top example, COSRA successfully identifies a `couch` in a living room setting. The context-aware prompt incorporates known objects (`person`, `bowl`, `apple`) with their spatial locations, combined with the unknown object's visual characteristics (fuzzy texture, medium size, plain pattern, decorative functionality, fur material, gray color, rectangular shape, urban environment, furniture category). This rich contextual information enables the LLM to generate accurate candidate labels including `Couch`, `Sofa`, `Tumbler`, `Recliner`, `Armchair` with `couch` being the top prediction.

The bottom example shows COSRA correctly identifying a `cup` in a dining context. The scene contains multiple known objects (`spoon`, `bowl`) that provide contextual cues about the dining/eating environment. The unknown object's characteristics (smooth texture, medium size, branded pattern, contemporary age, ceramic material, yellow color, cylindrical shape, eating use case, container category) guide the LLM to generate appropriate candidates including `Cup`, `Mug`, `Tumbler`, `Goblet`, `Chalice`.

These examples validate that COSRA's context-aware characterization and entropy-driven reasoning enable accurate label generation even without iterative refinement, demonstrating the effectiveness of our foundational approach.

### H.2  ADDITIONAL ITERATIVE REFINEMENT EXAMPLES

We provide further qualitative results of COSRA's predictions beyond those shown in the main text. These examples (Figure 8) illustrate both successful initial predictions and cases where iterative

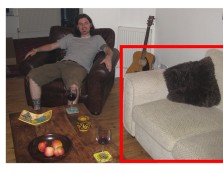

**Context-aware Prompt**: In a scene, we have **\*person\*** with bbox [111.6,119.1,328.6,353.7], **\*bowl\*** with bbox [130.4,402.0,236.9,474.7], **\*bowl\*** with bbox [246.4,358.4,282.6,396.4], **\*apple\*** with bbox [186.6,418.2,212.9,440.3]. And also have an unknown object with bbox [353.4,212.8,638.3,471.8] with these characteristics: textures: **Fuzzy, sizes: Medium, patterns: Plain, functionalities: Decorative, materials: Fur, colors: Gray, shapes: Rectangle, environments: Urban, categories: Furniture**. Based on the spatial relationships, context, and characteristics, what could this unknown object be? Give me a list of 50 possible objects for that place, ordered by likelihood.
**Answer**: < Couch, Sofa, Tumbler, Recliner, Armchair,…>

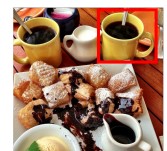

**Context-aware Prompt**: In a scene we have **\*spoon\*** with bbox [0.0,6.5,84.6,117.5], **\*spoon\*** with bbox [441.3,0.0,473.7,72.2], **\*bowl\*** with bbox [6.0,467.5,282.1,607.0]. And also have an unknown object with bbox [357.1,427.5,568.0,609.9] with these characteristics: **textures: Smooth, sizes: Medium, patterns: Branded, ages: Contemporary, materials: Ceramic, colors: Yellow, shapes: Cylindrical, use_cases: Eating, categories: Container.** Based on the spatial relationships, context, and characteristics, what could this unknown object be? Give me a list of 50 possible objects for that place ordered by likelihood.
**Answer**: < Cup, Mug, Tumbler, Goblet, Chalice,…>

**Figure 7: Examples of COSRA's immediate label generation capability.** Green boxes denote unknown objects that COSRA labeled correctly before refinement. The green text corresponds to the Attribute-Based Characterization (e.g., texture, size, color, etc.), which, together with scene context, guides the LLM to generate accurate candidate labels. Top: Living room scene with `couch` identification. Bottom: Kitchen scene with `cup` recognition.

refinement corrects early misclassifications. They highlight the robustness of COSRA across diverse object categories and scene contexts.

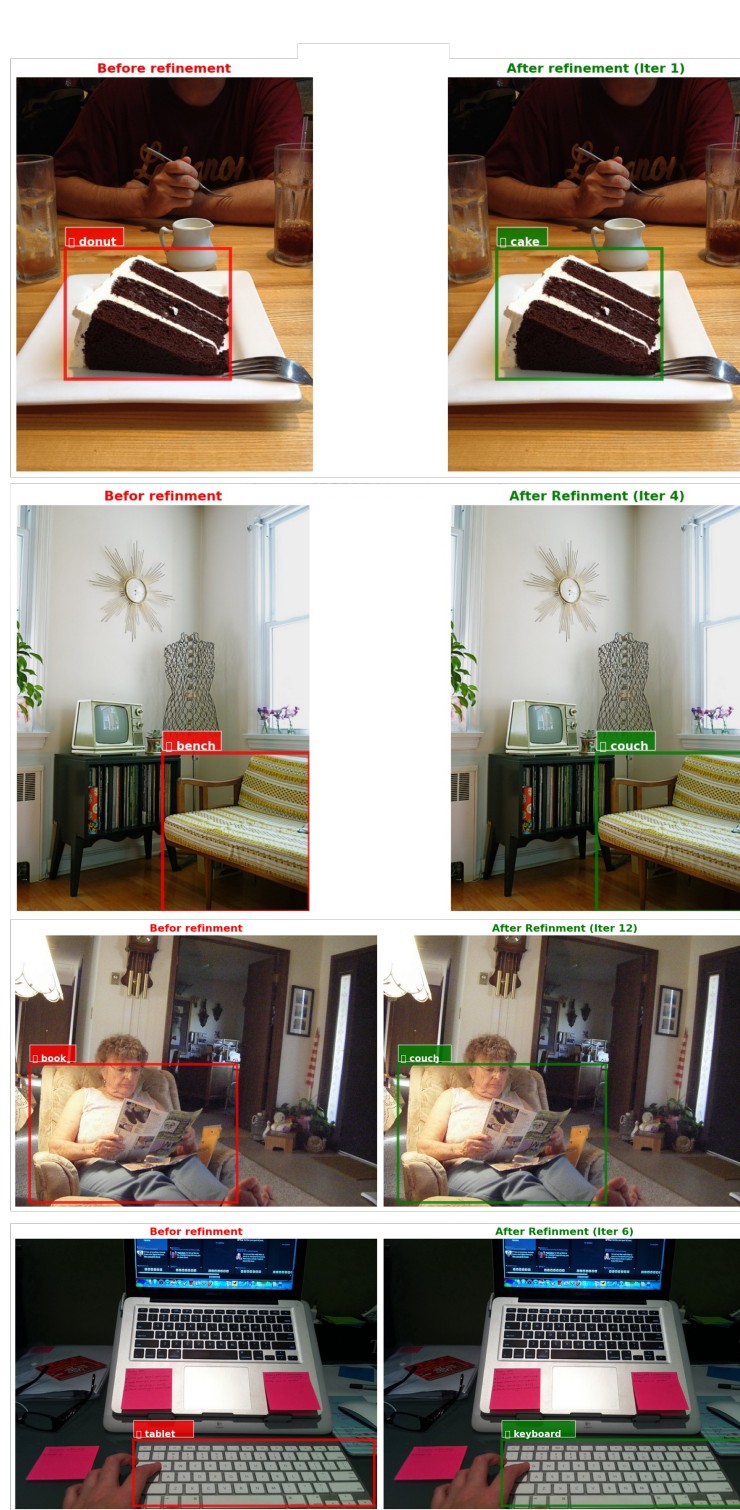

**Figure 8: Additional qualitative results of COSRA.** Examples show unknown objects and their predicted labels across refinement stages.

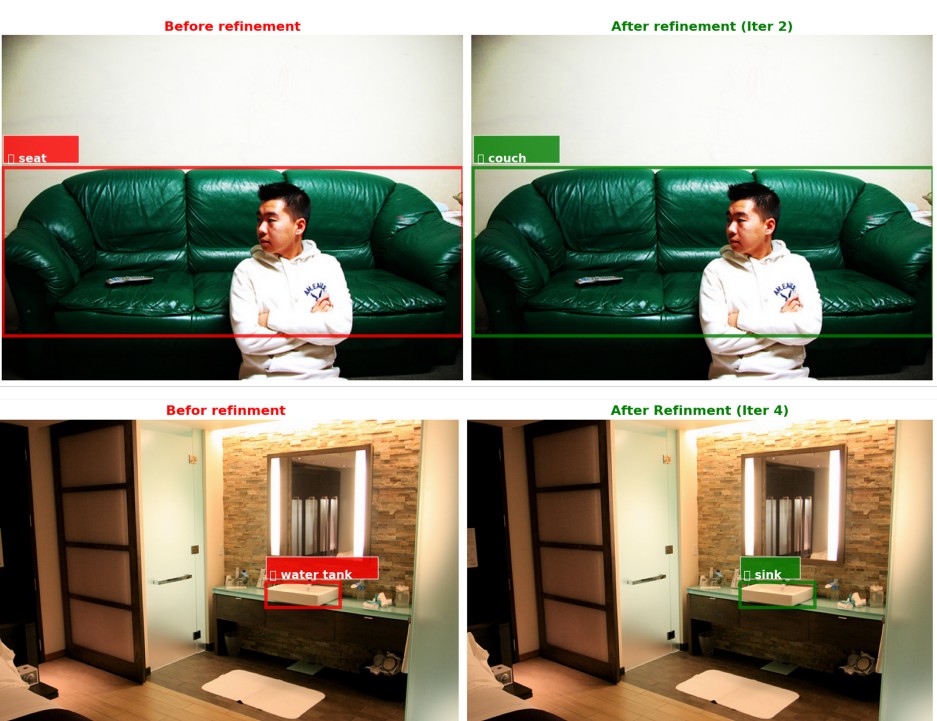

**Figure 8: Additional qualitative results of COSRA (continued).**

