# OpenReview forum: "Beyond Open-World: COSRA, a Training-Free Self-Refining Approach to Open-Ended Object Detection"
_ICLR.cc/2026/Conference — Submitted to ICLR 2026_

### Official Review · Reviewer_XkZN · 2025-10-27

**Soundness:** 2
**Presentation:** 2
**Contribution:** 2
**Rating:** 4
**Confidence:** 4

**Summary:**

The paper proposes COSRA, a training-free framework for open-ended object detection that aims to automatically assign semantic labels (e.g., “faucet” instead of “unknown”) to detected unknown objects. The method combines SAM and Faster R-CNN to propose unknown regions, uses a large language model (LLM) to generate candidate labels based on scene context and visual attributes, and refines these labels through CLIP-based re-ranking and an iterative self-correction process over a dynamic knowledge base called the Embedding-Label Repository (ELR). This enables open-ended naming without fine-tuning or reliance on a predefined vocabulary.

**Strengths:**

First, COSRA requires no training on novel categories and relies only on off-the-shelf foundation models (e.g., SAM, CLIP, and LLMs), offering practical convenience for deployment.
Second, the use of structured context-aware prompts to guide LLM label generation, combined with the ELR and iterative refinement, yields numerical results on COCO and LVIS that surpass current methods.
Third, the paper emphasizes open-ended naming, the ability to generate semantic labels without a fixed vocabulary, which conceptually advances beyond traditional open-world object detection.

**Weaknesses:**

The paper addresses an interesting direction: moving beyond open-world detection by assigning meaningful names to unknown objects. However, the experimental design and practical utility suffer from several critical issues:

- Redundancy with LLM capabilities: The core labeling mechanism relies entirely on an LLM to generate semantic names for unknown regions, using visual and contextual cues. Yet foundation models (e.g., GPT-4V, LLaVA 4) already demonstrate strong zero-shot object detection and naming abilities directly from images. It is unclear why a complex pipeline (SAM + detector + ELR + KNN voting) is needed when an LLM could, in principle, process the full scene and output both bounding boxes and labels end-to-end.

- Poor practicality for real-world use: The pipeline depends on multiple large models (SAM, CLIP, VLM, LLM) and expensive operations such as KNN search over a large repository. This makes real-time inference infeasible and limits applicability in resource-constrained settings.

- Risk of evaluation bias and knowledge leakage: The evaluation uses LLM-generated labels for unknown categories on standard benchmarks like COCO and LVIS. However, LLMs may exploit prior knowledge of dataset structure—for example, inferring likely “unknown” classes from the set of “known” classes. As shown in [1], providing GPT-3.5 with known COCO class names enables it to predict plausible unknown categories with high recall (~79%), nearly matching performance when the true unknown labels are given. This suggests that reported gains may stem from dataset-specific priors rather than genuine generalization.

- Insufficient reproducibility: Although pseudocode is provided, the overall pipeline is highly complex, involving prompt engineering, attribute extraction, ELR construction, and multi-stage refinement. Without full implementation details or code, it is difficult for readers to reproduce or build upon the method.

[1] Open World Object Detection in the Era of Foundation Models

**Questions:**

In light of the above concerns, the authors should address the following:

- Given that modern LLMs can perform open-ended detection directly, why not use an LLM as the primary detector? Please compare COSRA against end-to-end LLM-based detectors (e.g., on RefCOCO or similar grounding benchmarks) to justify the added complexity.
- Please report the computational cost (e.g., inference time, memory usage) of each component, especially the KNN search and LLM calls, to assess real-world feasibility.
- Please clarify whether the ELR construction or LLM prompting introduces dataset-specific bias or knowledge leakage, particularly given the strong prior knowledge of COCO/LVIS category structures in current LLMs.
- Please provide more core code, detailed descriptions of the prompt templates, attribute extraction, and ELR update logic to ensure reproducibility.

---

> ### Author Response · Authors · 2025-11-21
>
> ## W1/Q1: LLMs and MLLMs for Object Detection
>
> We thank the reviewer for their comments and feedback.
> Recent work has shown that multimodal LLMs perform poorly as open-ended object detectors because they are fundamentally designed for different tasks. LMM-Det evaluates LLaVA-style models on COCO and reports near-zero AP. These models perform much better on datasets like RefCOCO, where the task is to localize a specific target based on a provided textual description. However, RefCOCO is the opposite of our problem: it assumes the label is known and localizes the object, whereas we localize unknown regions first and then propose and refine candidate labels.
>
> Even when integrated into object detection frameworks, MLLMs do not always perform well. For example, CogVLM achieves 92.44% on RefCOCO, but when integrated into VL-SAM, the resulting method achieves 23.4 AP on LVIS rare categories, which COSRA outperforms. While using MLLMs within larger frameworks is a valid approach, our context-aware, self-refining pipeline and verification steps are designed to improve upon this strategy and already outperform methods that rely on MLLMs.
>
> ## W2/Q2: Computational Cost and Practicality
>
> COSRA is designed for scenarios where accuracy is prioritized over speed, such as dataset annotation, safety-critical analysis, and rare-event detection. While not suitable for real-time detection, COSRA remains practical for many real-world applications.
>
> Although inference is slower, the total computational cost is far lower than competing models because COSRA requires no training. Below, we summarize the computational requirements of related methods:
>
> ### Training and Inference Comparison
>
> | Method        | Training Hardware | Training Time | Training Data       | Inference             |
> |---------------|-------------------|----------------|----------------------|------------------------|
> | GenerateU     | 8× V100           | 4 weeks        | VG + GRIT (5M)       | Fast                   |
> | DetCLIPv3     | 32–64× V100       | Not reported   | 50M+ images          | Fast                   |
> | COSRA (Ours)  | None              | 0              | None                 | 3.4s per object*       |
> | (Runs on 3× RTX 3090) |           |                |                      |                        |
>
> \* LLM labeling takes 3.4 seconds per object. KNN refinement takes 38 minutes for 12 iterations over approximately 90k objects.
>
> ## W3/Q3: Evaluation Bias and Knowledge Leakage
>
> For LLM prompts and ELR construction, we do not provide any class label list as in open-vocabulary detection. We only provide detected known-class objects as scene context. We ensure the language remains generic and never indicates that known classes come from a specific dataset. Examples of the exact prompts are shown in Figure 7 of the appendix.
>
> The only potential source of leakage arises from the LLM inferring unknown objects based on co-occurrence patterns with known classes. However, such contextual inference also appears in realistic settings (e.g., tables co-occur with chairs both in COCO and real-world environments). This limitation is shared by all LLM or MLLM-based methods using the COCO base/novel split. On LVIS, this risk is reduced due to its long-tail distribution, so more weight may be placed on LVIS results.
>
> We use COCO and LVIS primarily because they are standard benchmarks for open-ended detection. As more robust datasets emerge, they can better serve as benchmarks for future work.
>
> ## W4/Q4: Reproducibility
>
> The full implementation code for COSRA will be released upon publication to ensure full reproducibility.
>
> ## References
>
> [1] LLMDet: Learning Strong Open-Vocabulary Object Detectors under the Supervision of Large Language Models, CVPR 2025
> [2] CogVLM: Visual Expert for Pretrained Language Models, arXiv 2023
> [3] VL-SAM: Training-Free Open-Ended Object Detection and Segmentation via Attention as Prompts, NeurIPS 2024
> [4] GenerateU: Generative Region-Language Pretraining for Open-Ended Object Detection, CVPR 2024
> [5] DetCLIPv3: Towards Versatile Generative Open-Vocabulary Object Detection, CVPR 2024

---

### Official Review · Reviewer_cEFM · 2025-10-27

**Soundness:** 3
**Presentation:** 2
**Contribution:** 3
**Rating:** 4
**Confidence:** 3

**Summary:**

This paper proposes COSRA, a training-free framework for open-ended object detection that combines context-aware reasoning from an LLM with CLIP-based visual alignment. The method iteratively refines object hypotheses and labels through self-learning without requiring fine-tuning or category supervision. The problem is highly relevant, and the approach is conceptually novel, aiming to move beyond conventional open-vocabulary detection.

**Strengths:**

1. Novel problem setting: The paper addresses open-ended object detection, an underexplored but crucial challenge in open-world vision.
2. Innovative idea: The “training-free + self-refining” approach that integrates LLM reasoning and CLIP similarity is original and well-motivated.
3. Thorough experimentation: Includes comparisons with diverse baselines (closed-set, open-vocabulary, and open-ended models) and well-structured ablation studies.
4. Clarity and presentation: The paper is clearly written, and the design of the “Embedding–Label Repository” and refinement loop is easy to follow.

**Weaknesses:**

1. Overstated “training-free” claim: The method heavily depends on pretrained foundation models , making performance and reproducibility strongly tied to these models.
2. Limited robustness and validation: The system assumes semantic similarity corresponds to visual similarity, which may not hold across object categories. There is no objective metric for evaluating the correctness of generated labels.
3. Potential circularity: CLIP similarity is used both for candidate selection and evaluation, introducing possible bias or circular validation.
Missing analysis: The impact of different LLM/VLM choices and LLM hallucinations is not explored.
4. Scalability concerns: The approach may face efficiency and cost issues in large-scale or real-time detection due to reliance on LLM inference.

**Questions:**

1. On the “training-free” claim:
Could the authors clarify the boundary of “training-free”? Since COSRA relies on pretrained SAM, CLIP, and LLaMA, to what extent can the system’s performance be considered independent of prior training? It would help to explicitly discuss whether COSRA’s novelty lies in architecture design or knowledge transfer from these pretrained models.

2. LLM dependence and hallucination:
How does the system handle cases where the LLM generates incorrect, irrelevant, or hallucinated object labels? Are there mechanisms (e.g., confidence filtering, cross-checking) to ensure semantic validity of generated labels? Quantitative or qualitative examples would strengthen the pape

---

> ### Author Response · Authors · 2025-11-21
>
> ## W1: Training-Free Definition
> We thank the reviewer for their comments and feedback. Training-free means COSRA performs no gradient-based training or fine-tuning: the detector, SAM, CLIP, and LLM/VLM remain frozen. The only updates occur in the ELR via non-parametric k-NN refinement.As with VL-SAM, GenerateU, and Open-Det, COSRA benefits from foundation-model pretraining. Our contribution is the training-free pipeline that: (i) extracts unknown regions via known-object detection + SAM, (ii) builds context-aware prompts, (iii) re-ranks with CLIP, and (iv) applies a self-refining ELR. The novelty lies in this inference design and in showing it can match or exceed training-based methods on novel classes.
> ## W2: Robustness and Validation
> COSRA does not assume a strict semantic–visual match. It only assumes same-category objects form loose clusters in vision–language space and relies primarily on contextual cues (co-occurrence, layout, interactions).
> COSRA integrates (i) region embeddings, (ii) LLM label candidates, (iii) CLIP image–text scores, and (iv) an ELR grouping instances by visual proximity + label similarity, all conditioned on scene context. When semantics/visuals disagree, these items appear as contextual outliers and are corrected by k-NN. Table 3 shows removing scene context causes the largest drop, confirming context—not local similarity—is the dominant robustness signal.Iterative ELR refinement consistently improves Novel/Rare AP, with performance increasing as the ELR grows. If assumptions were brittle, refinement would amplify noise; instead, it stabilizes predictions. For evaluation, COSRA’s free-form labels are mapped to COCO/LVIS via a fixed CLIP-based synonym-expanded protocol identical for all methods.
> ## W3
> ### Potential circularity (CLIP in method and evaluation)
> This design follows prior open-ended detection work such as GenerateU and Open-Det, which also use a frozen CLIP text encoder both inside the method and for mapping generated labels to dataset categories. In COSRA, CLIP is likewise a frozen, off-the-shelf component with two roles: (i) re-ranking LLM candidates and filtering attributes, and (ii) mapping free-form labels to COCO/LVIS classes in evaluation. The evaluation protocol is identical for COSRA and all baselines, so any CLIP bias affects all methods equally. Our ablations keep CLIP and the evaluator fixed and show that removing COSRA components or altering hyperparameters produces large drops in Novel and Rare AP—evidence that gains come from the context-aware, self-refining pipeline rather than from any circular use of CLIP.
>
> ### LLM/VLM choices
> Our focus is pipeline design. Representative LLM/VLMs are fixed to avoid confounds. COSRA is modular: any LLM generating label candidates and any VLM providing image–text similarity can replace current components in prompting, re-ranking, and refinement.
> ### LLM hallucinations
> A large part of COSRA is explicitly designed to mitigate LLM hallucinations:
> (i) attributes are filtered by a high image–text similarity threshold,(ii) LLM candidates are re-ranked by CLIP on the object crop,
> (iii) the ELR groups instances into visual–semantic clusters and applies iterative k-NN refinement.
> Hallucinated or inconsistent labels rarely form dense, visually coherent clusters and are thus unstable under refinement.
> ## W4: Scalability and LLM Cost
> Querying an LLM for every region is impractical for real-time use. COSRA is intended for offline annotation and novel-category discovery, where seconds per image are acceptable.
>
> COSRA queries the LLM only for unknown regions; known categories bypass the language pipeline and serve only as scene context. A fixed number \(m\) of name candidates is generated per unknown object, and ELR + k-NN refinement reuses labels across similar instances, reducing marginal cost as the repository grows.
>
> COSRA’s computational regime matches other training-free open-ended detectors (VL-SAM, GenerateU, Open-Det), all of which rely on large LLM/VLM inference; COSRA eliminates training cost entirely. As shown in Table 9, increasing \(m\) yields a clear accuracy–cost trade-off.
>
> ## Q1: Boundary of Training-Free Methods
> Training-free means no gradient updates; all backbone models remain frozen, and ELR refinement is non-parametric. COSRA’s novelty lies in combining complementary pretrained signals: detector regions, SAM boundaries, LLM contextual cues, and CLIP visual–semantic consistency. Ablations confirm complementarity: removing scene context, CLIP re-ranking, or ELR refinement causes large drops in Novel/Rare AP, showing each component is necessary.
> ## Q2: How LLM Hallucinations Are Handled
> COSRA counters hallucinations through attribute filtering, CLIP re-ranking, and ELR clustering with iterative k-NN refinement, ensuring visual–semantic consistency and stability across instances. Figure 6 illustrates a hallucinated label corrected through refinement.

---

### Official Review · Reviewer_b61E · 2025-10-29

**Soundness:** 1
**Presentation:** 3
**Contribution:** 1
**Rating:** 2
**Confidence:** 4

**Summary:**

This paper proposes a training-free framework to address open-ended object detection through context-aware reasoning and self-refining mechanisms. The pipeline contains three steps: In the first step, it uses SAM as a class-agnostic region proposal generator. And it classifies proposals into known and unknown subsets. In the second step, it first uses CLIP to assign some predefined attributes to unknown objects and then uses an LLM to generate free-form classnames for them based on the context. Finally, the method refines the predictions via grouping. The model is tested on COCO and LVIS and gets relatively high performance.

**Strengths:**

1. The paper is well-written. Although the proposed method is a little bit complex, this article presents the entire method in a highly organized manner, enabling readers to clearly understand the whole pipeline. Further, the red boxes in equations make them easy to understand.
2. The paper provides an in-depth analysis in Section 4.3, showing that the method is scalable with the repository size and scene context. These analyses provide a deeper understanding of the method.

**Weaknesses:**

1. The claim in Line 057-059 ''no existing system fully solves open-ended naming cleanly—i.e., given a truly unseen object, produce a correct semantic label (e.g., “giraffe”) without that name ever appearing in its vocabulary'' is overclaimed.
- The proposed method uses CLIP for similarity comparison, which is pretrained on massive image-text pairs. And novel object classnames exist in CLIP's vocabulary.
- The proposed method also uses an LLM to generate open-ended classnames. The LLM is pretrained on massive object description texts so that it can predict the classnames based on the context.
2. As the paper uses a strong LLM to predict classnames, why not directly use a strong MLLM (like Qwen3-VL[1], DAM[2]) to predict the classnames directly from pixels? These models already have strong region perception ability.
3. The comparisons with other methods in Table1 and Table2 are not fair, as the proposed method uses other strong models like SAM and LLaMA-4.
4. The comparisons in Table1 and Table2 lack many strong baselines, including but not limited to [3,4,5,6].
5. As shown in Table 9, the inference speed is significantly slower than the others.

I am quite sorry. But currently, the paradigm does not make sense to me.




[1] Qwen3-VL. https://huggingface.co/Qwen/Qwen3-VL-8B-Instruct

[2] Describe Anything: Detailed Localized Image and Video Captioning. In ICCV 2025.

[3] DetCLIPv3: Towards Versatile Generative Open-vocabulary Object Detection. In CVPR 2024.

[4] T-Rex2: Towards Generic Object Detection via Text-Visual Prompt Synergy. In ECCV 2024.

[5] LLMDet: Learning Strong Open-Vocabulary Object Detectors under the Supervision of Large Language Models. In CVPR 2025.

[6] A Hierarchical Semantic Distillation Framework for Open-Vocabulary Object Detection. In TMM 2025.

**Questions:**

Please see the weaknesses above.

---

> ### Author Response · Authors · 2025-11-21
>
> ## W1: Claim About Open-Ended Naming
> We thank the reviewer for their comments and feedback. We acknowledge the concern about overstated advantages, as other recent works in open-ended detection have also made progress. We will revise the wording for the camera-ready version.
> Our initial description may have been confusing. Our intention was to highlight a gap: CLIP and LLMs, when used independently, cannot fully leverage their vocabularies for end-to-end object naming. CLIP can only match an image to labels explicitly provided by the user; it cannot generate new names. LLMs can generate names, but without region-level visual grounding they cannot ensure that the generated label corresponds to the actual visual content. As a result, neither model alone can consistently produce correct names for truly unseen objects. Our method addresses this gap by combining visual grounding with open-ended label generation.
>
> ## W2: Why Not MLLMs Directly?
>
> Empirical evidence shows that multimodal LLMs perform poorly as object detectors. LMM-Det reports near-zero AP on COCO for LLaVA-style models, mostly due to extremely low bounding-box recall even after supervision. This highlights that MLLMs lack the spatial priors needed for accurate localization.
> Other works, such as VL-SAM, also used MLLMs (for example, CogVLM-17B) to query present objects and obtain textual responses. However, even with an MLLM, VL-SAM still required SAM for precise detection and achieved 23.4 AP on LVIS rare categories. These results suggest that MLLMs are inherently stronger at language understanding than at precise visual localization. While integrating MLLMs within a broader framework is valid, our context-aware, self-refining design improves upon this strategy and outperforms other methods that rely heavily on MLLMs.
>
> ## W3: Comparison Fairness and Missing Baselines
> We appreciate the reviewer’s concern about comparison fairness. Using foundation models such as SAM, CLIP, LLMs, and MLLMs is standard practice in current open-ended and open-vocabulary detection literature.
>
> Many OED and OVD approaches rely on such pretrained models, including:
> - VL-SAM, which uses both SAM and an MLLM.
> - GenerateU, which uses MLLMs and text-only LLMs for naming and refinement.
> - Open-Det, which uses an LLM as its core reasoning module.
> - DetCLIPv3, T-Rex2, and LLMDet, all of which depend heavily on CLIP or MLLM backbones.
>
> Thus, our use of SAM and LLaMA-4 aligns with established practice and with the baselines in Tables 1 and 2.
>
> COSRA also remains training-free, similar to VL-SAM: CLIP is frozen, SAM is off-the-shelf, and no detector or classifier is trained on COCO or LVIS categories. Most other baselines (except VL-SAM) require training on novel classes, pseudo-labels, or distillation. Therefore, COSRA is evaluated in a more challenging setting, while competing methods benefit from task-specific training. This matches standard practice in OED/OVD comparisons.
>
> ## W4: Missing Baselines
> We were not aware of these baselines at submission time, which is why they were not included. Methods such as DetCLIP-v3, T-Rex2, and LLMDet mainly improve training-based open-vocabulary detectors through large-scale fine-tuning, supervision, or distillation. In contrast, COSRA is training-free and focuses on grounded open-ended naming, generating labels for unseen concepts rather than classifying within a fixed vocabulary. Even when training-based methods achieve strong results, COSRA offers complementary advantages in flexibility and operation without retraining or predefined label sets.
>
> ## W5: Inference Speed
> COSRA is designed for scenarios where accuracy is prioritized over speed, such as dataset annotation, safety-critical analysis, and rare-event detection. Although not suitable for real-time use, COSRA remains effective for many offline applications.
>
> While inference is slower, the overall computational cost is far lower than training-based methods because COSRA requires no training. Inference can also be optimized through quantization and batching to reduce memory and LLM overhead.
>
> ### Training and Inference Comparison
>
> | Method        | Training Hardware | Training Time | Training Data | Inference Speed     |
> |---------------|-------------------|---------------|---------------|----------------------|
> | GenerateU     | 8× V100           | 4 weeks       | VG+GRIT (5M)  | Fast                 |
> | DetCLIPv3     | 32–64× V100       | Not reported  | 50M+ images   | Fast                 |
> | COSRA (Ours)  | None (3×3090)     | 0             | None          | 3.4s per object*     |
>
> *LLM labeling: 3.4s/object; ELR refinement ~38 minutes for 12 iterations over ~90k objects.
>
> ## References
> [1] LLMDet, CVPR 2025
> [2] VL-SAM, NeurIPS 2024
> [3] CogVLM, arXiv 2023
> [4] GenerateU, CVPR 2024
> [5] Open-Det, ICML 2025
> [6] DetCLIPv3, CVPR 2024
> [7] T-Rex2, ECCV 2024

---

> ### Comment · Reviewer_b61E · 2025-11-26
> **Reply to the comment**
>
> Thanks authors for providing rebuttal. However, the low performance and the high latency make the paper not convincing.
>
> Regarding comment W2: MLLMs are poor at localization but excel at recognition. This characteristic is shared by LLMs and it is not a reason to use CLIP+LLMs for recognition instead of MLLMs.

---

> > ### Author Response · Authors · 2025-12-03
> >
> > ## Additional Clarification on Performance, Latency, and MLLMs
> >
> > ### On performance and latency.
> >
> > It is unclear to us why the reviewer thinks our model has low performance. Against other existing open-ended detection methods, COSRA is the highest performing model to our knowledge. Additionally, COSRA outperforms many top-performing open-vocabulary models while solving the more difficult task of open-ended detection (see Tables 1 and 2). In this more constrained setting, COSRA:
> > (i) achieves competitive AP on COCO and LVIS;
> > (ii) substantially outperforms prior training-free systems such as VL-SAM on novel/rare categories; and
> > (iii) corrects hallucinated or incorrect labels via iterative refinement (see Figure 2 in the main paper and Figures 6 and 8 in the supplementary).
> >
> > As for latency, many tasks prioritize accuracy over speed, such as offline labeling of large image sets, and as COSRA was not designed for real-time use its value should be assessed accordingly. Another consideration is the model development cost, as COSRA trades a few seconds per image for the ability to annotate unseen categories without retraining. In contrast, training-based methods pay a large *up-front* cost (weeks of multi-GPU training), which COSRA entirely avoids. From a total-compute perspective, COSRA offers tradeoffs from a design standpoint: zero training cost, a modular framework, slightly higher per-image inference cost, and strong open-ended naming ability.
> >
> > ### On W2: Why CLIP + LLM instead of a single MLLM?
> >
> > We agree that MLLM's poor localization ability is not sufficient to entirely disregard MLLMs as a potential method. However, just because MLLMs are a valid approach, that does not mean that using CLIP+LLM is not also viable. Other recent models also use a VLM+LLM approach (LLMDet, DVDet), reporting strong performance, showing that VLM+LLM can be a very competitive option. We also highlight that the approach of using an MLLM for recognition has already been explored by methods such as VL-SAM and GenerateU, and COSRA outperforms both (by 3.7% for VL-SAM and 7.1% for GenerateU on $AP_{rare}$ for LVIS minival).
> >
> > **Moreover, current evidence does not fully support the claim that modern MLLMs excel at visually grounded recognition**: for example, HallusionBench reports that GPT-4V attains only 31.42% question-pair accuracy, with all other evaluated LVLMs below 16% [1], revealing persistent issues with context-induced hallucinations and conflicts between linguistic priors and actual image content. This is unsurprising, as large language models are primarily optimized for *linguistic* competence, and their multimodal variants often amplify language-driven biases rather than enforcing pixel-level grounding. Thus, we chose to implement a CLIP+LLM approach for our work.
> >
> >
> > References:
> > [1] HallusionBench: Guan et al., “HallusionBench: An Advanced Diagnostic Suite for Entangled Language
> > Hallucination and Visual Illusion in Large Vision-Language Models,” CVPR 2024.

---

### Official Review · Reviewer_QQMA · 2025-11-01

**Soundness:** 3
**Presentation:** 3
**Contribution:** 2
**Rating:** 4
**Confidence:** 3

**Summary:**

This paper tackles the challenging task of open-ended object detection, where the system must assign correct semantic labels to truly unseen objects (e.g., identifying a “giraffe” without ever having that label in its vocabulary). The proposed approach, COSRA (Context-oriented Open-ended Self-Refining Annotation), is a training-free framework that leverages context-based reasoning and self-refinement. Experiments on COCO and LVIS demonstrate strong performance, suggesting the effectiveness of the method in open-world recognition settings.

**Strengths:**

The core idea is clear and well-motivated, and the example provided in the Introduction effectively illustrates the problem setting.

The paper is well-written and the exposition makes it easy to follow the main contributions.

The proposed self-refinement strategy by combining iterative visual-cohesion reasoning with K-NN-based relabeling, whihc offers a principled approach to mitigating label noise.

**Weaknesses:**

Although presented as a training-free approach, the method relies heavily on multiple computationally expensive pre-trained models (e.g., SAM, Faster R-CNN, CLIP, LLaMA), which raises questions about practicality and fairness in comparison to other methods.

The selection of system components is not sufficiently justified. For example, the use of Faster R-CNN rather than DETR for object detection is not explained, and the claim that “any detector can be employed” is not empirically supported.

The write-up claims SOTA on COCO/LVIS but lacks detailed comparisons to strong baselines.

**Questions:**

What are the memory and resource figures?

Provide failure cases illustrating scenarios where COSRA struggles.

---

> ### Author Response · Authors · 2025-11-21
>
> ## W1: Practicality and Fairness Concerns
> We appreciate the reviewer's concern and agree that pretrained models such as SAM, Faster R-CNN, CLIP, and LLaMA are computationally heavy to train. Our use of them follows the established definition of "training-free" adopted in recent works such as VL-SAM [1] and BLIP-2 [2]: the method performs no task-specific training or fine-tuning. While these models are large to train, they are inexpensive to use, and our pipeline adds zero additional training cost. Our framework is designed for scenarios where accuracy is prioritized over speed, such as dataset annotation, safety-critical analysis, and rare-event detection. despite not supporting real-time use, it can still benefit many real-world applications.
> Regarding fairness, recent work in OED/OVD increasingly relies on powerful pretrained vision and language models. For example, VL-SAM combines SAM with an MLLM in a way that is closely aligned with our use of SAM and LLaMA-4. GenerateU [6] incorporates both multimodal and text-only LLMs for naming and refinement, while Open-Det centers its entire pipeline around an LLM-based reasoning module. Likewise, many strong OVD baselines—including DetCLIPv3 [3], T-Rex2 [4], and LLMDet [5]—depend heavily on pretrained CLIP or MLLM backbones. In light of this broader landscape, the concern that our comparisons are "not fair" simply because COSRA uses SAM and an LLM does not reflect current practice; these components are widely adopted across the very methods used as baselines in Tables 1 and 2.
> ## W2: Component Selection Justification
> We use Faster R-CNN because its proposal-based design offers high recall and reliable region coverage, which is important for capturing potential unknown regions. While we adopt Faster R-CNN in our implementation, the framework is detector-agnostic: any detector producing bounding boxes can be substituted, as we do not depend on detector-specific logits or vocabularies. The pipeline is modular, and the choice of detector does not influence any of the downstream components.
> The other components serve general functional roles: segmentation for masks, vision–language models for semantic similarity, and LLMs for contextual reasoning. These choices are practical rather than restrictive, and each module can be replaced by alternatives with similar capabilities, keeping the framework modular and flexible.
> ## W3: Missing Strong Baselines
> We were not aware of these baselines at submission time, which is why they were not included in our comparisons. Nonetheless, we agree that discussing the differences is important. Methods such as DetCLIP-v3 [3], T-Rex2 [4], and LLMDet [5] focus on improving training-based open-vocabulary detectors through large-scale fine-tuning, supervision, or distillation. In contrast, our approach is training-free and targets grounded open-ended naming, where the system must generate labels for unseen concepts rather than classify from a predefined vocabulary. Even if training-based OVD methods report strong results, our method offers complementary advantages in flexibility and the ability to operate without retraining or fixed label sets.
> ## Q1: Memory and Resource Figures
>
> COSRA is designed for scenarios where accuracy is prioritized over speed, such as dataset annotation, safety-critical analysis, and rare-event detection. While not suitable for real-time detection, COSRA can still benefit many real-world applications.
>
> We also highlight that while our inference is slower, the total computational cost is orders of magnitude lower than other models due to zero training requirements. We provide runtime information and comparisons in the table below:
>
> | Method | Training Hardware | Training Time | Training Data | Inference |
> |--------|------------------|---------------|---------------|-----------|
> | GenerateU [6] | 8× V100 | 4 weeks | VG+GRIT (5M) | Fast |
> | DetCLIPv3 [3] | 32-64× V100 | Not reported | 50M+ images | Fast |
> | **COSRA** | **None (3× RTX 3090)** | **0** | **None** | **3.4s/obj**\* |
> \*LLM labeling: 3.4s/object; Refinement: 38 min for 12 iterations over ~90K objects
> ## Q2: Failure Cases
> Thank you for the suggestion. We agree that presenting failure cases would strengthen the paper. In the camera-ready version, we will include representative examples where COSRA struggles.
>
> ## References
> [1] VL-SAM: Training-Free Open-Ended Object Detection and Segmentation via Attention as Prompts. In NeurIPS 2024.
>
> [2] BLIP-2: Bootstrapping Language-Image Pre-training with Frozen Image Encoders and Large Language Models. In ICML 2023.
>
> [3] DetCLIPv3: Towards Versatile Generative Open-vocabulary Object Detection. In CVPR 2024.
>
> [4] T-Rex2: Towards Generic Object Detection via Text-Visual Prompt Synergy. In ECCV 2024.
>
> [5] LLMDet: Learning Strong Open-Vocabulary Object Detectors under the Supervision of Large Language Models. In CVPR 2025.
>
> [6] GenerateU: Generative Region-Language Pretraining for Open-Ended Object Detection. In CVPR 2024.

---

> > ### Comment · Reviewer_QQMA · 2025-11-27
> >
> > Thank you for the detailed response. I am still unclear about the idea of “using SAM and Faster R-CNN to separate known from novel objects,” especially in the case of real novel objects.
> >
> > Because SAM may generate masks for object parts rather than whole objects, how should we handle these incorrect or partial candidate unknowns? Additionally, Faster R-CNN can struggle with out-of-distribution data and may produce overconfident predictions for unseen categories. Since the goal is to detect novel objects, what happens when the novel objects are difficult even for SAM to segment?

---

> > > ### Author Response · Authors · 2025-12-03
> > >
> > > We appreciate the reviewer’s follow-up and clarify that Stage 1 is a *high-recall proposal filter*, not a perfect oracle for “real” novel objects. SAM first produces dense, class-agnostic region proposals using 32×32 grid sampling with strict quality thresholds (Table 7), pruning very small or unstable fragments while keeping broad coverage. A Faster R-CNN trained only on base classes then scores these regions, and we use it *conservatively*: a region is marked as known only if its base-class score is very high; otherwise it is kept as a candidate unknown. Thus, the detector is used only to “claim” clearly base-class objects, while ambiguous or out-of-distribution regions are intentionally routed to the unknown branch rather than filtered out. Regarding SAM producing part-level masks, the later stages of COSRA are explicitly designed to absorb such noise. Part segments or imperfect masks can still be useful as long as they contain discriminative visual evidence; very small or inconsistent regions tend to (i) fail the attribute similarity threshold tau_attr and contribute weak textual cues, and/or (ii) appear as visual outliers during ELR refinement, where they are relabeled via k-NN voting or effectively down-weighted. Appendix G and Figures 2 and 6 show that initial mistakes (e.g., *sponge* vs. *sink*, *motorcycle* vs. *couch*) are corrected once enough visually similar neighbors are present, indicating that noisy or partial proposals are repaired by the refinement loop rather than dominating predictions.
> > >
> > > If the detector is not clearly confident about a box, we do *not* mark it as known — we send it to the unknown set. Also, SAM produces its own class-agnostic masks, so even when the detector mislabels a region, overlapping SAM masks that are not confidently known still go into ELR.
> > >
> > > If a novel object is genuinely hard for SAM to segment (i.e., no reasonable proposal is produced), it falls outside the recall of any SAM-based method. This limitation is shared by open-world and open-vocabulary detectors that rely on region proposals, including VL-SAM and GenerateU+SAM variants. Nevertheless, the strong Novel/Rare AP we obtain on COCO and LVIS suggests that, for these benchmarks, SAM + Faster R-CNN provide sufficiently rich candidate regions in practice. The ELR and its refinement stages then handle the inevitable mix of whole-object, part-level, and noisy proposals. We also emphasize that COSRA’s main contribution is the *structure* of the context-aware naming and self-refining ELR pipeline. As discussed in the paper, Faster R-CNN and SAM are not fixed components and can be replaced by any detector and class-agnostic proposal generator with adequate recall.
> > >
> > > We hope this clarification helps address the reviewer’s concern.

---

### Meta-Review · Area_Chair_5Jh5 · 2026-01-07

**Summary:**

Reviewers expressed concerns about the paper's novelty (overclaiming open-ended naming), practical utility (heavy reliance on pre-trained models leading to high computational cost and slow inference), fairness of comparisons (missing strong baselines, potential evaluation bias), and reproducibility.

**Reviewer Concerns:**

Addressed:
- Clarified training-free definition, justified component selection, and provided resource figures and failure case plans.
- Acknowledged overclaim, explained why not use MLLMs directly, and argued comparisons are standard.
- Defined training-free boundary, discussed robustness mechanisms, and addressed circularity.
- Explained why not end-to-end LLM, compared computational cost, and promised code release.

Outstanding:
- Practicality concerns may persist due to computational expense, but rebuttal improved clarity.
- Reviewer remains unconvinced about performance and latency; core issues unresolved.
- Scalability concerns and LLM hallucination handling not fully resolved.
- Practicality and reproducibility depend on future code release; bias risks may remain.

**Reviewer Scores:**

- Reviewer QQMA: Score likely unchanged at 4, as practicality concerns may not be fully alleviated.
- Reviewer b61E: Score unchanged at 2, as reviewer was not convinced by rebuttal.
- Reviewer cEFM: Score might increase to 6 if clarifications on robustness satisfy, but could stay at 4.
- Reviewer XkZN: Score might increase to 6 with code release.

---

### Decision · Program_Chairs · 2026-01-26

Reject